# Identifying global hotspots of agricultural expansion into non-forest ecosystems

Siyi Kan[1,2], Jing Meng [1] ✉, U. Martin Persson [3], Bin Chen [4], Samuel A. Levy [5], Elise Mazur[6], Leah Samberg[5], Guoqian Chen [7], Heran Zheng [1] & Thomas Kastner [2]

Ecologically important non-forest ecosystems, including grasslands, shrublands and wetlands, face substantial threats from agricultural expansion, yet their conversion dynamics remain poorly understood. This study identifies global hotspots of land conversion from non-forest (and forest) ecosystems to cultivated lands from 2000 to 2020, including conversion within Protected Areas and its impacts for biodiversity conservation. Using three state-of-the-art land cover datasets (GlobeLand30, GLCLUC and GLC_FCS30D), we find extensive and increasing non-forest conversion, often comparable to or exceeding forest conversion. Protected non-forest ecosystems cover substantially smaller area than protected forests while experiencing disproportionately high conversion rates. Non-forest and forest conversion together affected habitats of over 5,000 threatened species, over half of which depend critically on non-forest ecosystems. Our study provides important insights for improved land cover data development, while offering companies and policymakers science-based evidence to design sustainable land-use policies and integrated policy frameworks that avoid trade-offs and support broad sustainability goals.

Agricultural expansion has led to extensive loss of natural ecosystems, consequently affecting terrestrial biodiversity and carbon stocks[1,2]. Numerous studies have assessed the status of deforestation and forest degradation[3–5]—primarily in the tropics—but a narrow focus on forests is far from enough to support global biodiversity and climate goals. Non-forest natural ecosystems, including natural grasslands, shrublands, and wetlands, are of great ecological importance globally. Grasslands store over one-third of global terrestrial carbon and have the potential to sequester 2.6–8.0 billion tons of carbon dioxide equivalents per year through soil organic carbon storage[6,7]. They also hold over 33% of global biodiversity hotspots and contain habitats of global importance[8]. Due to their greater resilience to extreme heatwaves, drought, and wildfire compared to forests, they are considered

as more stable and reliable carbon sinks in areas where forests are climate-vulnerable[9]. As reported by the Convention on Wetlands and previous literature, wetlands harbor 40% of all known animal and plant species (including numerous endemic species)[10] and hold over 30% of the world's soil organic carbon[11], despite covering only 6% of the global land surface.

Although a few local-scale studies have shown that non-forest ecosystems are critically imperiled, most existing literature have focused either on mapping distribution of non-forest lands[12,13] or on analyzing forest conversion[3–5]. To date, however, no study has comprehensively assessed the global dynamics and impacts of non-forest ecosystem conversion (beyond static distribution), and particularly, the role of agricultural expansion in driving these changes, which

[1]The Bartlett School of Sustainable Construction, University College London, London, UK. [2]Senckenberg Biodiversity and Climate Research Centre (SBIK-F), Frankfurt am Main, Germany. [3]Department of Space, Earth & Environment, Physical Resource Theory, Chalmers University of Technology, Gothenburg, Sweden. [4]Fudan Tyndall Center, Department of Environmental Science & Engineering, Fudan University, Shanghai, China. [5]Rainforest Alliance, New York, NY, USA. [6]Land & Carbon Lab, World Resources Institute, Washington, DC, USA. [7]Laboratory of Systems Ecology and Sustainability Science, College of Engineering, Peking Univinsity, Beijing, China. ✉e-mail: jing.j.meng@ucl.ac.uk

represents an urgent gap that must be addressed to inform sustainable land use policies. For instance, almost half of the Cerrado's native vegetation has been converted, at a rate higher than deforestation in the Amazon[14], and forest-focused policies risk causing unintended leakage to non-forest ecosystems, thereby compromising their intended environmental benefits[15–17]. Meanwhile, there is growing interest in expanding supply chain governance to protect non-forest ecosystems, such as through no-conversion commitments. For example, the Accountability Framework Initiative and the Science Based Targets Network advocate for no-conversion policies that include non-forest ecosystems[18]. Numerous companies have also pledged to reduce their impact on all terrestrial ecosystems beyond just avoiding deforestation, including several of the 500 companies that are most influential driving tropical ecosystem loss[19]. These actions require precise knowledge of where non-forest conversion takes place, yet there is currently a lack of products and databases providing such information. In addition, a global analysis on Protected Areas (PAs) finds that, in contrast to forested PAs, non-forest PAs across the tropics face higher human pressures than similar non-protected areas[20], but the extent of land conversions within non-forest PAs remains unknown.

Understanding is also limited regarding the impacts of non-forest ecosystem conversion on biodiversity, which hampers public policymakers' and companies' ability to design and implement effective biodiversity conservation policies. Agricultural expansion and management is one of the primary drivers of biodiversity loss[21], a topic that has received renewed global focus in light of the Kunming–Montreal Biodiversity Framework proposed at the UN Biodiversity Conference. Neglecting the conversion of non-forest ecosystems and associated impacts on biodiversity could potentially misguide policy deployment and jeopardize the timely achievement of conservation objectives. Although existing literature has studied various dimensions of biodiversity loss[22–24], disentangling the impacts that the conversion of different land cover types have can help reveal additional gaps in ongoing conservation strategies.

In light of these challenges, this analysis utilized 30 m-resolution geospatial datasets to investigate spatiotemporal patterns of global agricultural expansion into various natural land covers from 2000 to 2020, as well as the occurrence of conversion within PAs and potential impacts on terrestrial biodiversity. Emphasis was placed on non-forest land covers, with forests examined for comparative purposes. Global total conversion and conversion within PAs were estimated by overlapping maps of land use and land covers[25–27] and PAs[28]. Biodiversity impacts were evaluated from three complementary perspectives: species richness in converted areas, threatened species whose habitats overlapped with converted areas, and conversion within Key Biodiversity Areas (KBAs), by integrating maps of land conversion, species' ranges[29,30], KBAs[31], and data on species' habitat preferences[29].

To minimize potential misestimation from relying on a single data source, the analyses were conducted using three state-of-the-art land cover datasets: GlobeLand30[25], Global Land Cover and Land Use Change (GLCLUC)[26], and GLC_FCS30D[27]. These datasets, each compiled with distinct methodologies and different land cover classification schemes (see Supplementary Information (SI) Table S1), provide complementary information that together enable a more holistic and robust assessment of land conversion. For example, GLC_FCS30D applies a change detection algorithm that enhances accuracy in identifying pixels of change[32] and it is the only dataset that includes tree crops—a major driver of agricultural expansion in Southeast Asia and Africa[33,34]—in its cropland class. Only GlobeLand30 tries to differentiate cultivated pastures from natural grasslands based on a rigorous 3-step method, which is important given the large share of grazing in agricultural land use[35]. GLCLUC, however, uses 4 years of data for the identification of cropland to minimize misclassifications (e.g., due to fallow). Therefore, the comparison across datasets allows us to account for methodological limitations and misclassifications of the

underlying remote sensing products datasets, as well as ensure that the conversion dynamics captured by the analysis are robust.

All three datasets consistently confirm extensive and increasing agricultural encroachment on global non-forest ecosystems and identify a common set of hotspot countries. By identifying global conversion hotspots, our study can assist companies and public policymakers in the design and prioritization of sustainable land-use policies, such as future conservation planning and conversion-free supply chain policies. Moreover, our quantitative global-scale assessment of conversion rates as well as cross-dataset comparisons not only provide a starting point for future improved quantification of land use transitions and related eco-environmental impacts, but also advances our understanding of the limitations of currently available datasets and challenges for land conversion assessments, offering valuable insights where improved data are most needed to support more robust analyses.

## Results

### Status of natural land conversion

The figures calculated in this study should be interpreted with a clear understanding of the underlying land cover classifications (Table 1). For simplicity, the terms Grassland and Shrubland, Non-forested Wetland, and Forestland are used to represent the corresponding classes in the GlobeLand30, GLCLUC, and CLC_FCS30D datasets, and they are collectively referred to as natural land covers, including semi-natural land covers that may be managed but still possess many characteristics of natural areas (e.g., managed grasslands). Accordingly, agriculture-driven non-forest conversion refers to the conversion from grasslands, shrublands, and non-forested wetlands to cultivated lands (in GlobeLand30) or croplands (in GLCLUC and GLC_FCS30D). However, it is important to note that the definitions and scopes of these terms vary across the datasets (see SI Table S1 for more details). Conversion is assessed by comparing land cover classifications in 2000 and 2010 for the period 2000–2010, and in 2010 and 2020 for the period 2010–2020.

At a global scale, non-forest natural land covers spanned 3.0–5.5 Gha in 2000, while forest areas covered 4.1–5.1 Gha. Non-forest conversion driven by the expansion of cultivated lands/croplands increased between 2000 and 2020, being as widespread as or more pervasive than forest conversion (Fig. 1b). The variations across the three datasets for non-forest conversion is much smaller than that for forest conversion. About 75–109 Mha of grasslands and shrublands were converted during 2000–2010, increasing to 92–120 Mha during 2010–2020. The conversion of non-forested wetlands ranged between 2 and 7 Mha and 2 and 6 Mha in the two decades, respectively. Overall, non-forest conversion amounted to 173–243 Mha during 2000–2020, compared to 18–173 Mha of forest conversion. Despite the variations across the three datasets, they show high geographical overlap in the identified conversion hotspots (Fig. 1a and SI Fig. S1). The temporal trend of conversion at the pixel level is shown in SI Fig. S2.

Figure 1c shows hotspot countries for non-forest conversion with the highest average conversion areas across the three datasets and identified in at least two of them. Five countries consistently appeared in the top ten hotspots across all three datasets: Brazil (22–32 Mha), China (11–19 Mha), Russia (8–17 Mha), the United States (10–23 Mha), and Argentina (5–9 Mha). Seven additional countries/economies are identified in any two of the datasets: Australia (7–15 Mha), India (2–21 Mha), Nigeria (3–17 Mha), the EU (2–12 Mha), Kazakhstan (4–11 Mha), Tanzania (3–7 Mha), and Mexico (3–6 Mha). They altogether made up about 61%–66% of global total non-forest conversion.

Variations across the three datasets can be partly explained by their different classification schemes for agricultural lands and highlights the importance of selecting an appropriate dataset for regional analyses based on local circumstances. GlobeLand30 is the only dataset potentially capturing the important impact of pasture

**Table 1 | Land cover classifications in this study and corresponding classes in each dataset**

| This study | Cultivated land/Cropland | Grassland & Shrubland | Non-Forested Wetland | Forestland |
|---|---|---|---|---|
| GlobeLand30 | Cultivated land (Exclude tree crops, Include cultivated grassland) | Grassland and shrubland | Wetland | Forestland |
| GLCLUC | Cropland (Exclude tree crops, Exclude cultivated grassland) | Short vegetation and tree cover <5 m in Terra Firma class | Short vegetation and tree cover <5 m in Wetland class and open surface water present 20–79% of the year | Tree cover ≥ 5 m, incl. forested wetlands |
| GLC_FCS30D | Cropland (Include tree crops, Exclude cultivated grassland) | Grassland and shrubland | Marsh, salt marsh, flooded flat, tidal flat, and saline in Wetland class | Forestland and swamp and mangrove in Wetland class |

**Note:** GLCLUC's classification is mainly based on land cover canopy (for short vegetation cover) and tree height (for tree cover). Following FAO's definition that forests should have trees higher than 5 m, tree cover ≥ 5 m in GLCLUC dataset was compared to forestland in the other datasets, while short vegetation and tree cover < 5 m in its Terra Firma and Wetland classes were compared to grassland & shrubland and non-forested wetland, respectively. Similarly, GLC_FCS30D's swamp and mangrove (both covered with woody vegetation) were classified as forestland, which provides a conservative estimate of non-forested wetlands as some woody land covers are not forests. Since this distinction was not possible for GlobeLand30 and non-forested wetland area was 1.5 times larger than forested wetland area in 2000 (according to GLC_FCS30D), the entire wetland class in Globeland30 was excluded from the forestland category.

expansion, by including cultivated pastures in its cultivated land class. This may be one reason why GlobeLand30 reports much larger agriculture-induced non-forest conversion in countries such as Brazil, Australia, and Nigeria, where pastures represent a large share of agricultural land use[35]. GLC_FCS30D can account for the influence of tree crops by including them in the cropland class, e.g., in Indonesia, where oil palm is a major cause of deforestation[33] (SI Fig. S4). GLCLUC defines cropland as land used for herbaceous crops, and this study attributes trees <5 m to short vegetation following FAO's definition of forests[36]. Therefore, GLCLUC and GLC_FCS30D could misestimate conversion of natural land vegetations when they transitioned to cultivated pastures or when cultivated pastures (and short tree crops for GLCLUC) were converted to croplands.

Different classifications of natural land covers also contribute to the observed discrepancies. For instance, GLC_FCS30D sets lower tree canopy threshold for forests than GlobeLand30, and neither GlobeLand30 nor GLC_FCS30D specify whether tree height was a criterion for forests, while GLCLUC does not consider tree canopy information, both key criteria for FAO's definition of forests. Meanwhile, GlobeLand30 and GLC_FCS30D classify Tundra as a separate category but use different classification standards, whereas GLCLUC includes it within tree cover and short vegetation. As a result, GLCLUC shows the largest non-forest area, and GLC_FCS30D reports the largest forest area globally, which can further lead to discrepancies in the observed conversion area. Wetlands are generally defined as lands transitional between terrestrial and aquatic systems, but only GLCLUC provides a clear quantitative identification criterion. Besides, GLCLUC and GLC_FCS30D contain several wetland subcategories, enabling classification of forested wetlands under forestland and allowing a specific focus on non-forested ecosystems. In contrast, GlobeLand30 does not support this distinction, necessitating the exclusion of its entire wetland class from forestland. Notably, total wetland conversion in GlobeLand30 was smaller than non-forested wetland conversion in the other two datasets, indicating differences in classification standards. Further research is still required to disentangle the extent to which the observed discrepancies result from differences in definitions versus inaccuracies in the underlying data sources.

## Conversion within protected areas

PAs covering natural non-forest ecosystems occupy a substantially smaller area than those protecting forests (SI Fig. S3), partly attributable to the longstanding emphasis of forests in global conservation[17,37]. However, the conversion area of protected non-forest land covers to cultivated lands/croplands (4–6 Mha) was comparable to the conversion of protected forestlands (1–8 Mha, Fig. 1b). Meanwhile, all three datasets implied increasing conversion within PAs for both forest and non-forest ecosystems over time.

Figure 2a illustrates the distribution of total converted PAs across different PA management categories defined by the International Union for Conservation of Nature (IUCN)[38]. Most conversion of protected grasslands and shrublands took place in categories IV (Conservation through active management), V (Landscape conservation and recreation), and VI (Sustainable use of natural resources), though the strictly protected category II (Ecosystem conservation and protection) also accounted for around 14–20% of the total. Protected non-forested wetlands show a similar pattern, with more conversion in category IV compared to protected grasslands and shrublands.

Figure 2b compares conversion rates inside PAs by IUCN PA categories to that outside PAs at the country level (i.e., conversion inside PAs/total PA area vs. conversion outside PAs/total non-PA area). We focus on the pattern for the entire period, therefore, the conversion rates were calculated as the share, per country, of PAs established before 2000 that were converted between 2000 and 2020. Overall, conversion rates for non-forest land covers were higher than those for forestlands and were higher outside than inside PAs. While global average and median conversion rates for different PA categories were typically below 10%, some countries exhibited higher rates. For 96% (GLC_FCS30D) to 99% (GLCLUC) of the approximately 180 countries having PAs in 2000, conversion rates remained below 60%. In countries exhibiting high conversion rates for a specific PA category, the total PA area in that category is typically small. The high rates may result from limited enforcement of agricultural restrictions in those regions (e.g., in Cambodia[39]), or from dataset misclassifications (as even misclassifications of a few pixels within these PAs can lead to substantial differences in conversion rates), which may explain why no country with a conversion rate exceeding 60% appeared consistently in at least two datasets. In general, GLCLUC and GlobeLand30 datasets show similar patterns at a global scale: conversion rates increased with the PA category number from categories I (Strict Protection) to V, while category V showed higher rates than category VI. GLC_FCS30D generally reveals much higher conversion rates, even in strictly controlled categories I & II, which possibly indicates that tree crops were more likely to encroach on protected areas. Nevertheless, these differences cannot be used directly to infer conclusions about PA effectiveness, as land inside and outside PAs may have very different characteristics, e.g., accessibility or agricultural suitability[40]. Future in-depth causal research is needed to examine the causes of these patterns.

## Impacts on terrestrial biodiversity

Figure 3a provides an overview of species richness where natural land conversion has occurred, depicting the number of species potentially occurring in each grid cell. We included amphibians, mammals, reptiles, and birds listed as threatened species (categorized as Vulnerable (VU), Endangered (EN), and Critically Endangered (CR)) according to IUCN Red List Categories. The highest concentrations of threatened species were mainly found in South and Southeast Asia, which means even a small area of conversion in these regions might have great biodiversity impacts.

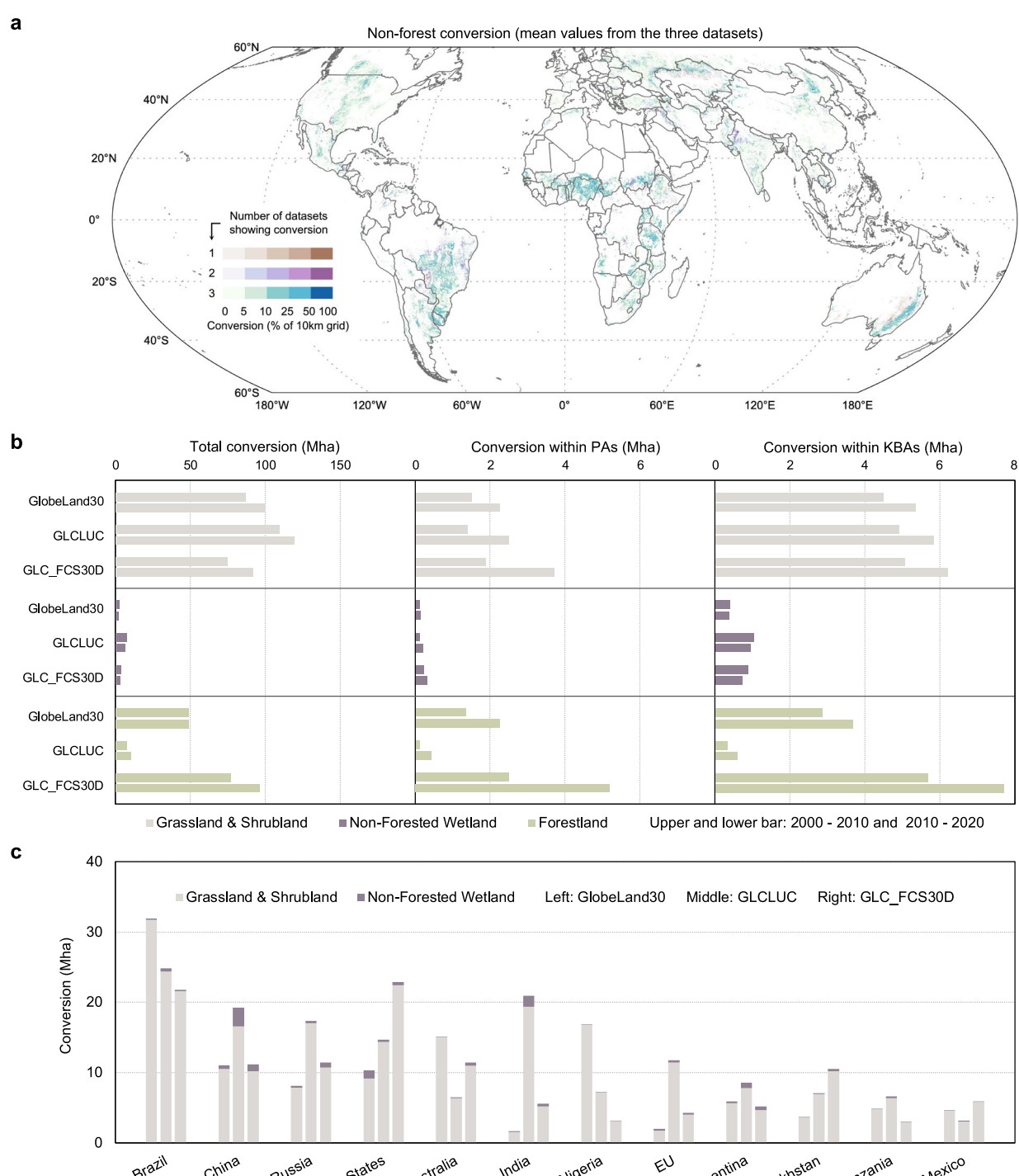

**Fig. 1 | Conversion of natural land covers to cultivated lands/croplands.**
a Global distribution of non-forest conversion hotspots in 2000–2020. Values (mean from the three datasets) are presented as the percentage of 30 m by 30 m pixels in a 10 km by 10 km grid cell for which such a conversion has been identified. The colored areas account for 90% of global total non-forest conversion; b Comparisons across datasets regarding total conversion, conversion within

Protected Areas (PAs), and conversion within Key Biodiversity Areas (KBAs); c Non-forest conversion for the 5 hotspot countries/regions appeared in all 3 datasets and the 7 hotspot countries/regions appeared in any 2 of the datasets. The European Union (EU) is analyzed as an aggregate. Administrative boundaries were modified from GADM version 4.1, aggregated, and simplified for display purposes.

Figure 3b analyzes affected species by taxonomic group and habitat affinity. Overall, habitats of over 5000 threatened species overlapped with areas of non-forest or forest land cover conversion. Habitats of over 1700 amphibians were affected, followed by reptiles, mammals, and birds (each ranging between 1000 and 1200). More threatened species endemic to forests were potentially affected than

those endemic to grassland & shrubland or wetland ecosystems (2310 vs. 625). Nevertheless, over 55% of the affected species are not endemic to forests and rely on non-forest ecosystems for their habitats. Notably, non-forested wetlands, despite their small land area, are home to over 1500 affected threatened species, whether they are endemic or not. Due to the high workload of analyzing all the datasets at high

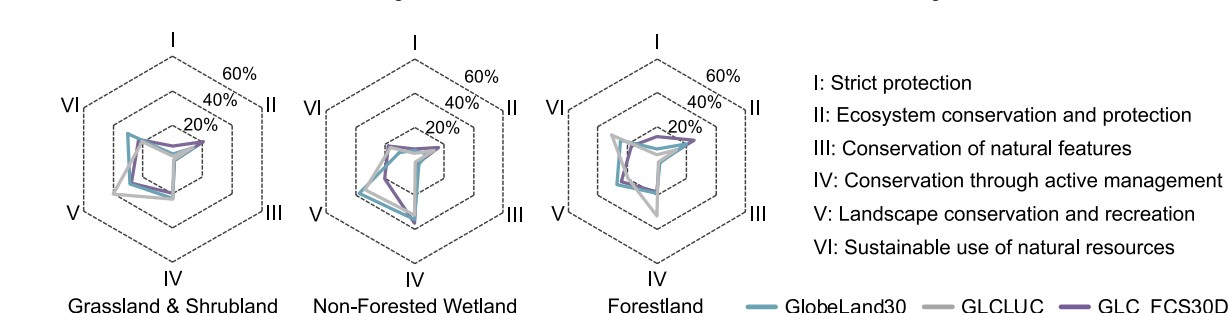

**a** Distribution of global conversion within PAs across different IUCN PA categories

I: Strict protection
II: Ecosystem conservation and protection
III: Conservation of natural features
IV: Conservation through active management
V: Landscape conservation and recreation
VI: Sustainable use of natural resources

Grassland & Shrubland   Non-Forested Wetland   Forestland

— GlobeLand30   — GLCLUC   — GLC_FCS30D

**b** National conversion rates inside and outside PAs

GlobeLand30

GLCLUC

GLC_FCS30D

**Fig. 2 | Conversion within protected areas (PAs) per country. a** Proportion of conversion within each IUCN Protected Area (PA) category relative to total conversion across all PA categories. **b** Percentage of conversion inside PAs (area converted within PAs/total PA area) by IUCN category, compared with conversion outside PAs (area converted outside PAs/total non-PA area, shown as No). Each column represents a specific IUCN PA category and the associated land cover type within PAs in 2000, summarizing conversion rates across countries. Boxes show the interquartile range, with the lower and upper boundaries representing the first and third quartiles, respectively, and horizontal lines inside indicating the median. Whiskers extend to the minimum and maximum values within 1.5 times the interquartile range, and dots mark outliers beyond this range. For clarity, vertical axes are capped at 60%, covering 96% (GLC_FCS30D) to 99% (GLCLUC) of ~180 countries with PAs in 2000.

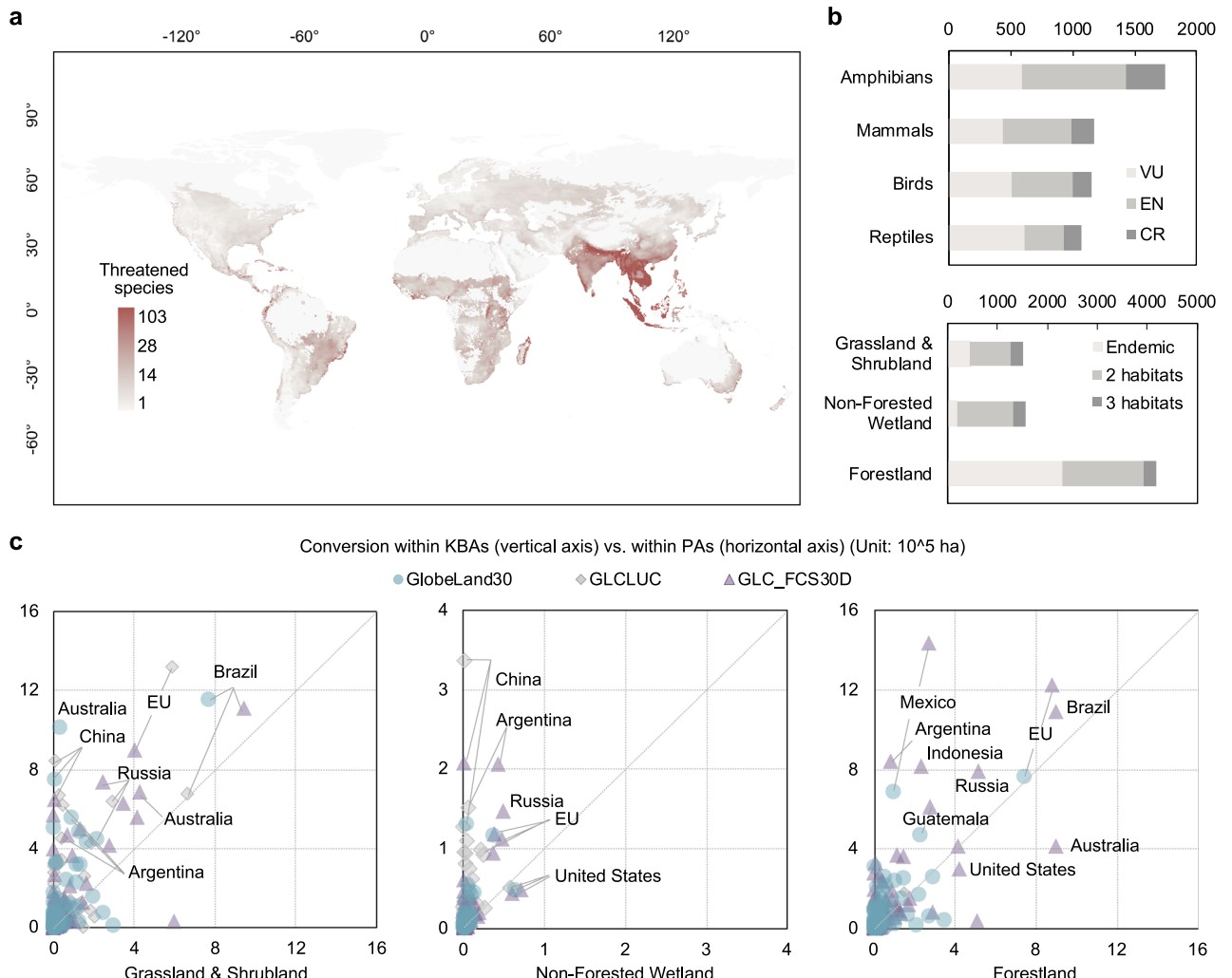

**Fig. 3 | Influence of conversion on terrestrial biodiversity. a** Richness distribution of threatened species in grids detecting natural land conversion (including both non-forest and forest conversion) across all three land cover datasets. Species richness was assessed and displayed at -30 km resolution; **b** Number of threatened species whose habitats overlapped with areas of natural land conversion, categorized by taxonomic group (upper panel) and habitat affinity (lower panel). Three habitats indicates species with affinity to grassland & shrubland, non-forested wetland, and forestland; Two indicates affinity to the ecosystem shown on the vertical axis and one other ecosystem; and endemic indicates species restricted to the corresponding ecosystem; **c** Conversion within Key Biodiversity Areas (KBAs) and comparison with conversion within Protected Areas (PAs). Land boundaries were modified from GADM version 4.1, aggregated, and simplified for display purposes.

spatial resolution, GlobeLand30 was used to identify these affected threatened species, as it includes cultivated pastures, which occupy a large proportion of total agricultural land[41]. However, the impact of excluding the other two datasets is likely minimal, because only a few areas show conversion in those datasets but not in GlobeLand30 (see Fig. 1a and SI Fig. S1). Additionally, species habitats usually extend over large areas, beyond individual pixels where differences between datasets occur, and nearby areas were consistently identified as converted across all three datasets, suggesting that affected species were likely already accounted for in the analysis.

Figure 3c explores conversion within KBAs, which are the most crucial sites supporting substantial populations of global threatened species. KBAs are identified by applying a series of criteria and thresholds to reveal important dimensions rather than just numbers of species (e.g., geographically restricted biodiversity, ecological integrity, and habitats irreplaceability)[31]. Globally, there was increasing conversion within KBAs, with non-forest conversion comprising 49%–93% of the global total (Fig. 1b). In comparison to conversion within PAs, it can be found that a large share of KBAs undergoing conversion were not protected through PAs, and the gap for non-forest ecosystems was larger than for forests. This

situation was particularly notable for grasslands and shrublands in Australia, China, the EU, Russia, and Argentina, as well as non-forested wetlands in China, Argentina, and the EU. Extensive conversion within KBAs also occurred for grasslands and shrublands in Brazil and non-forested wetlands in the United States, although the gap between converted PAs and KBAs in these regions was relatively smaller.

## Discussion

To design and implement policies for conserving natural non-forest ecosystems, it is imperative to uncover the extent and temporal trend of their conversion. Our results provide a global spatiotemporal analysis of agriculture-driven conversion of natural non-forest ecosystems. This information can be used by policymakers and companies that produce or source agricultural commodities to assess the implications of priority conservation areas, no-conversion supply chain policies, and other sustainable land-use strategies.

Extensive non-forest conversion to cultivated lands/croplands underscores the threatened state of non-forest ecosystems and a sizable gap in existing conservation efforts. The urgency is further emphasized by findings indicating that non-forest conversion impacts

biodiversity at a scale similar to forest conversion. Although the establishment of PAs has yielded some positive outcomes, the coverage of PAs for non-forest ecosystems remains inadequate, compared to the targets established under the Kunming–Montreal Biodiversity Framework, leaving higher proportions of non-forest ecosystems unprotected. Even areas already under protection continued to experience increasing conversion during the study period, and substantial KBAs undergoing conversion were not protected through PA networks. These findings highlight the need to further scale up the coverage and enforcement of PAs and other conservation actions. Moreover, future land management policies should consider all local ecosystem types, as policies narrowly targeted at one ecosystem could risk displacing production to other ecosystems with weaker legal protections[17,42]. In addition to existing conversion hotspots, it is also essential to pay more attention to non-forested tundra and permafrost regions. These areas store vast amounts of carbon but are increasingly threatened by global warming, which causes thawing and makes them more accessible to human activity[43]. Accordingly, more interdisciplinary studies are needed to project the impacts of climate change on these vulnerable regions.

All three datasets identified Brazil, the United States, China, Russia, and Argentina as common hotspots of non-forest conversion. Additionally, Australia, the EU, India, Kazakhstan, Mexico, Nigeria, and Tanzania appeared as conversion hotspots in at least two of the datasets. In Brazil, our findings are in line with numerous recent analyses[44–46], illuminating the need to expand public and private protections for natural ecosystems outside forest biomes, such as the Cerrado, Pampas, and Caatinga. Other regions we identify that should be prioritized for conservation include tropical and subtropical savannahs in Africa (particularly in Nigeria and Tanzania), temperate grasslands in south-eastern Australia, the Mongolian–Manchurian grasslands (especially within China), the Great Northern Plains in North America (notably in the United States), and the Eurasian Steppe (especially within Russia). Several countries identified as hotspots have implemented or are in the process of enacting conservation policies. For instance, Nigeria has committed to achieving Land Degradation Neutrality goals, which include taking actions to mitigate grassland decline by 2030[47]. The United States has authorized the Grassland Conservation Reservation Reserve Program[48] and Protection of Wetlands (Executive Order 11990)[49]. China has implemented a series of nationwide grassland and wetland conservation programs and legislations[50,51]. There are also recent or upcoming action plans, including Brazil's recent decree to expand the scope of public action to prevent conversion in all of its ecoregions[52]. Despite these initiatives, the scale of conversion indicates the need to intensify protection efforts. Our results can provide evidence-based insights about future priority conservation areas.

The spatial distribution of conversion hotspots is influenced by both geographic and socio-economic factors. Key geographic determinants include the extent and spatial distribution of agricultural and natural non-forest lands. Major conversion hotspots, such as Brazil, Russia, China, and the United States, also have the largest non-forest natural land area. However, Canada ranks only around 15th in non-forest conversion, despite ranking around 5th in non-forest natural land area. This is primarily due to Canada experiencing agricultural land contraction in recent decades[53]. Additionally, agricultural areas in Canada are primarily concentrated in the Prairies due to agricultural suitability, not located in the areas of the major remaining non-forest ecosystems. The localized distribution further minimizes the interface between cropland and adjacent natural areas, where agricultural expansion typically occurs[54,55], and increases the difficulty of accessing more remote natural areas. Socio-economic factors, such as policy regulations and environmental concerns, also play an important role[56]. Countries with less conservation policies and weaker enforcement tend to experience more conversion. For instance, Canada has 4.8

times more non-forest natural lands covered by PAs than Tanzania, but experienced far less conversion within these PAs. This can partly reflect its stronger conservation efforts and explain why its total non-forest conversion was lower than Tanzania, despite having over 3 times more croplands and 4.5 times more non-forest natural lands.

Our analyses also have important implications for supply chain actors, including companies and importer governments, which are found to drive natural ecosystem conversion both directly and indirectly through their demand for conversion-risk agricultural commodities and their sourcing or trade of these commodities from conversion hotspot countries[57]. Despite extensive non-forest conversion, most supply chain actions have been historically focused on forest protection. Although further research will be required to attribute the conversion to specific commodities or supply chains, many identified conversion hotspots, such as Brazil, the United States, Argentina, and Australia, are major producers and exporters of known conversion-risk agricultural commodities (e.g., soybeans)[35,57]. These findings already indicate that, companies and countries sourcing from these regions may be contributing to consequent biodiversity and carbon impacts, therefore, should adopt or extend their existing supply chain strategies to mitigate impacts across all ecosystems, not just forests. At the government level, the EU has proposal to extend the Regulation on Deforestation-free Products (EUDR) to include non-forest ecosystems[58], while broader international efforts remain needed.

Additionally, our attempt at the global-scale quantification of non-forest conversion as well as comprehensive comparisons across different datasets, reveal several key challenges for land cover conversion assessments at the global level and identify priority areas where improved data will be especially helpful to identify non-forest ecosystem conversion:

(1) As noted in the introduction, our approach to identify and quantify conversion is based on existing land cover datasets, and we did not directly process satellite image with change detection algorithms and conduct sampling for area estimates. We have tried our best to reveal the risks for spurious changes, including comparisons across different datasets, and comparisons between conversion to agricultural land and reversion from agricultural land to natural vegetation (SI Fig. S5), which can be a combined results of actual land use dynamics (e.g., cropland fallow, cropland abandonment, ecosystem restoration) and spurious changes from misclassification. We have also compared our results with existing work by Winkler et al.[59], which aligns with our findings in showing more extensive non-forest than forest conversion (SI Fig. S6). Remote sensing products are available that support change detection and sampling in forest ecosystems[60], and similar advances related to non-forest ecosystems are needed to improve the accuracy of non-forest conversion assessment.

(2) Similarly, technical challenges in remote sensing inevitably introduce some uncertainties in the development of land cover datasets, particularly those aiming for global coverage across all land cover types. While region-specific or single land cover-focused datasets may offer higher accuracy for local areas or specific land cover types, their limited scope makes them unsuitable for our globally consistent cross-country and cross-land cover comparisons. Future studies could benefit from improved land cover products tailored to specific research objectives. Until such products become widely available, incorporating local, context-specific knowledge remains essential for verifying and fully understanding land use dynamics.

(3) Our work also highlights the need for consistent definitions, classifications, and mapping of natural and cultivated ecosystems. To our best knowledge, no existing global fine-resolution land cover products distinguish cultivated pastures from natural

grasslands and tree crops from natural forests simultaneously, partly due to technical challenges. They also have different classification schemes for different natural land covers. This not only complicates quantitative comparisons between different analyses but also hinders the integration of various land cover and land use data (e.g., merging datasets compiled specifically for pastures with those for all types of land covers). More in-depth assessments and refinements should carefully select the most suitable dataset. For example, for some regions where non-forest ecosystems become prime targets not only for herbaceous cropland but also for tree crops and afforestation (e.g., in India[61]), datasets that distinguish tree crops (e.g., GLC_FCS30D) could be more suitable for conversion assessments.

(4)  Furthermore, more comprehensive analyses are necessary to assess issues related to PAs and biodiversity. Regarding PAs, this paper provides only preliminary estimation of conversion within PAs, but the effectiveness of PAs and spillover impacts of forest policies still need rigorous causal analyses. Regarding biodiversity, this research analyzed agricultural intrusion into species' habitats, nevertheless, the potential risks could go beyond the species directly affected by habitat loss due to interspecific relationships[62]. Evaluating biodiversity loss is also inherently intricate and challenging because of its multifaceted feature. Therefore, future research should specifically target each of the unexplored area on depth to provide a more comprehensive understanding.

## Methods

### Conversion of natural lands

We carried out a spatially explicit analysis of agricultural encroachment on natural ecosystems in order to reveal the distinct patterns of non-forest conversion versus forest conversion. Specific areas of conversion were identified by overlaying 30 m by 30 m land cover maps for the initial and final years of each period: the 2000 and 2010 maps for conversion between 2000 and 2010, and the 2010 and 2020 maps for conversion between 2010 and 2020. The overall area of conversion was then calculated by aggregating the area of each grid cell that showed conversion, at both the country scale and the 10 km scale (the Mollweide equal projection).

The approach to identify and quantify conversion is based on existing land cover products, and we did not directly process satellite image with change detection algorithms and conduct sampling for area estimates. To account for methodological limitations and misclassifications of the underlying remote sensing products datasets, as well as ensure that the conversion dynamics captured by the analysis is robust, we compared results based on three state-of-the-art fine-resolution land cover datasets—GlobeLand30[25], GLCLUC[26], GLC_FCS30D[27], which were created using different methods, notably, GLC_FCS30D used a change detection algorithm. Detailed explanation about limitations and challenges can be found in the Results and Discussion section of the main text. Detailed classification schemes are available in the SI Table S1.

### Conversion within protected areas (PAs)

The layers for natural land cover conversion and PAs are overlaid to evaluate conversion within PAs. The map for PAs was obtained from the World Database of Protected Areas[28]. We considered I–VI PA categories classified by IUCN, including I. Strict Nature Reserve and Wilderness Area, II. Ecosystem conservation and protection (i.e., National Park), III. Conservation of natural features (i.e., Natural Monument), IV. Conservation through active management (i.e., Habitat/Species Management Area), V. Landscape/seascape conservation and recreation (i.e., Protected Landscape/Seascape), and VI. Sustainable use of natural resources (i.e., Managed Resource Protected Area). Since the land cover conversion maps were generated on a decadal basis, we restricted our assessment of conversion within PAs to those PAs established before each respective time frame. However, this approach may underestimate encroachment, as newly established PAs during the examined period could also experience land cover conversion afterward. For the conversion rates inside and outside PAs, we focus on the pattern over the entire period concerned (i.e., 2000–2020). Therefore, we only considered PAs established before 2000 and calculated the share converted between 2000 and 2020.

### Impacts on terrestrial biodiversity

Affected species were estimated by overlaying land cover conversion maps with the range layers of each threatened species. We used range polygons of all the threatened amphibians, mammals, reptiles provided by IUCN[29], and birds provided by Birdlife International[30]. This paper focused exclusively on the species categorized as VU, EN, CR according to the IUCN Red List Categories. To analyze species richness, which is defined as the number of species potentially occurring in each grid cell, we followed procedures consistent with IUCN species richness assessments. The species ranges were intersected with the 865 km²-resolution IUCN ISEA10 hexagon grid that is considered more suitable for a range of ecological applications than rectangular grids, with each analysis exported to a 30 km-resolution raster in the Mollweide equal-area projection and reprojected to WGS84 for visualization. Richness was not analyzed at 30 m fine resolution because threats to biodiversity can extend beyond the immediate location of land cover conversion to vast surrounding areas due to edge effects and increased accessibility to remaining natural ecosystems facilitated by infrastructure development[63]. Meanwhile, focusing on small grid size can diminish the variation in species diversity between different regions. For instance, if species richness is assessed at a 30 m-resolution, two 900 km² areas—one sheltering a different species in each 900 m² grid and the other contains just one species throughout—could misleadingly have identical richness maps.

To estimate the number, taxonomic groups, and habitat affinities of species whose habitats overlapped with areas of conversion, the analysis was conducted using GlobeLand30 at fine resolution, but the impact of excluding the other two datasets is likely minimal, as explained in the Results section. The information of species' habitat affinity, for example, whether the species are endemic to grasslands or have habitats in both grasslands and wetlands, was also obtained from IUCN[29]. IUCN's forest classification includes subtropical/tropical mangrove vegetation above high tide level, subtropical/tropical swamp, and forests in other climate zones. Therefore, we infer that the IUCN reference to wetlands pertains specifically to non-forested wetlands.

In order to provide complementary insights beyond just numbers of species at risk, we also studied conversion within KBAs, based on BirdLife International KBA dataset[31]. KBAs are sites of exceptional biodiversity conservation value and are identified according to a series criteria and thresholds in a global standard. There are five broad categories of criteria, namely threatened biodiversity, geographically restricted biodiversity, ecological integrity, biological processes, and irreplaceability, which are further divided into specific sub-criteria. KBAs are identified based on scientific identification and shed light on critical locations for biodiversity conservation, while PAs are legally recognized and formal governed for various objectives, not solely out of biodiversity concern; therefore, assessing conversion within both KBAs and PAs offers a more comprehensive view of the challenges facing biodiversity conservation.

## Data availability

The land cover and use data used in this study are available in the GLCLUC database under accession code https://glad.umd.edu/dataset/GLCLUC2020, the GLC_FCS30D database under accession code https://doi.org/10.5281/zenodo.8239305 and the GlobeLand30 database under accession code https://www.webmap.cn/

mapDataAction.do?method=globalLandCover. The Protected Areas data used in this study are available in the World Database on Protected Areas under accession code https://www.protectedplanet.net/en/thematic-areas/wdpa. The Key Biodiversity Areas data used in this study are available in the World Database of KBAs under accession code https://www.keybiodiversityareas.org/request-gis-data. The data on species' ranges and habitat affinities are available in the IUCN Red List of Threatened Species under accession code https://www.iucnredlist.org/en. The administrative boundaries data used in this study are available in the GADM database under accession code https://gadm.org/download_world.html. The land conversion data generated in this study have been deposited in the Zenodo database under accession code https://doi.org/10.5281/zenodo.10909260.

## Code availability
The R codes used for data processing in this study have been deposited in the Zenodo database under accession code https://doi.org/10.5281/zenodo.10909260.

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

## Acknowledgements

We thank the Carbon Neutrality and Energy System Transformation programme (J.M.). We acknowledge the support by the National Key R&D Program of China under Grant no. 2023YFE0112902 and 2022YFE0209400 (J.M.), European Union under Grant no. 101137905 (PANTHEON, J.M.), the Research Grants Council of the Hong Kong Special Administrative Region, China under Grant no. AoE/P-601/23-N (J.M.), German Federal Ministry for Economic Cooperation and Development under Grant no. GS22 E1070-0060/029 (T.K. and S.K.), Gordon and Betty Moore Foundation under Grant no. 12194 (L.S. and S.A.L.), and the National Natural Science Foundation of China under Grant no. 52100210 (B.C.).

## Author contributions

S.K. conceived the study, conducted data processing, analyses, and visualization, and led the manuscript writing. T.K. contributed to the processing of the GLCLUC and GLC_FCS30D land cover data. U.M.P., S.A.L., E.M., L.S., and T.K. provided input on methods development. J.M., U.M.P., B.C., S.A.L., E.M., L.S., G.C., H.Z., and T.K. contributed to refining the study design and manuscript structure. All authors reviewed and edited the manuscript.

## Competing interests

The authors declare no competing interests.
