## [Transparent Peer Review File · Nature Communications]

Identifying global hotspots of agricultural expansion into non-forest ecosystems

Corresponding Author: Professor Jing Meng

Version 0:

Reviewer comments:

Reviewer #1

(Remarks to the Author)

Interesting study on a topic that does not receive much attention. There are two major points I would like to see addressed.

The first is to explain why Canada is not identified as a primary hotspot. Canada is the second largest country, by land mass, in the world. Canada has vast prairie grasslands. In addition, and perhaps more importantly, three of the largest wetlands in the world are found in Canada: The Hudson Bay Lowland, The Mackenzie River basin, and the prairie potholes of North America (shared with USA). Based on these facts it seems to me impossible that Canada would not be identified. And because of that, I question the data.

The second is to account for climate change. There is no mention of climate change in this study. There are vast regions of non-forested tundra and permafrost affected by climate change. There is interest in expanding agriculture into these areas as they become warmer. It would be valuable and interesting to forecast climate change effects on tundra and permafrost as they relate to agricultural expansion.

(Remarks on code availability)

Reviewer #2

(Remarks to the Author)

Thank you for an interesting publication. There is definitely a need to focus on non-forest ecosystems and your general conclusions are in line with other studies of this nature. However, I have some comments about the scope of the manuscript.

In the introduction, it would be useful to have slightly more emphasis (an explanatory paragraph) on why conservation of non-forest lands are important. This is necessary here because the arguments for a focus on forests have generally related to their unique richness. Grassland is, for instance, believed to be sometimes a better ecosystem for carbon storage than forests in areas where forests burn regularly, even though per area carbon storage is less.

I was looking for more discussion about how you are defining the three biomes considered and how the GlobalLand30 classification referred to has been interpreted in practice. "Wetland" is interpreted in very different ways around the world – the Convention on Wetlands includes coastal waters to 6 metres depth for instance, which I don't think you are including here, so it would be worth defining this clearly in the main body of the paper. Furthermore, I'm not sure the sources you are quoting in the introduction are necessarily using the same definitions. If peatlands are included in wetlands my guess is that the figure for 20% global soil carbon is an underestimate. Similarly, do your estimates cover the whole land surface, in which case where does tundra sit – within wetlands or grasslands (some but not all definitions of rangelands include tundra because of reindeer herding and use of caribou)? I think a statement is needed of what you understand has and has not been included in the various definitions (including differences between the three data sets, which you have tried to do).

While the general conclusions seem logical, i.e., the top-ranking countries for conversion, I'm confused by some of the figures, particularly related to protected areas. I assume from the caption on figure 2 that analysis is by country but find the reference to outlier data surprising. Are you saying that some countries have lost more than 90% of their non-forest

ecosystems inside protected areas over the last 20 years? If this is really the case this is dramatic news and the countries need to be identified. But I would be looking for a lot of verification with country experts before publishing this.

Similarly, while the caveats in the discussion section were very useful, they also throw question on some of the results. If existing methodologies cannot differentiate pasture from natural grassland (and this includes being unable to distinguish pasture resown with non-native species), or tree crops from natural forest, then what impact does that have on the results? I see that one of the three datasets used claims to include cultivated grassland amongst "cultivated land/cropland) but how is this done? Do you have opinions about the accuracy?

Finally, I had to read figure 2 several times to make sense of the figures, whereas figure S4 was immediately clear. I suggest including the latter in the main body of the paper.

(Remarks on code availability)

Reviewer #3

(Remarks to the Author)

This analysis was interesting, with great care taken to choose the land cover datasets and explain their benefits and limitations. The manuscript outlines the importance of non-forest systems for biodiversity and highlights the vulnerability of these systems (and the species that depend upon them) to conversion into cultivated lands (e.g., cropland). The results emphasize that the conversion of these lands has far outstripped that of forested lands over the last two decades, with rates of conversion increasing over time.

General comments:

Generally, I felt that the work was robust with a sound methodology, thanks to the use of multiple land cover datasets to find general patterns. However, I felt the link to policy - in particular, the idea of designing conversion-free supply chains - lacks some details and logic, leaving it feeling a little forced. I was surprised that Winkler et al. (2021) (<https://www.nature.com/articles/s41467-021-22702-2>) was not referenced; it would be helpful to compare the methods and results with this previous analysis, which also looked at land-use conversion rates over time. In some places, details on the results or methods are lacking, making it difficult to assess the overall importance of some of the take-home messages, but these issues could be easily resolved. I also would have appreciated a more detailed assessment of why the spatial patterns in conversion might be observed. The authors have made an effort to clearly comment their code, making it much easier to review.

Specific comments:

Abstract:

- This was well written and interesting, however, it is lacking in some key statistics to support the take-home messages. For instance "protected non-forest ecosystems cover substantially less area than protected forests, while their conversion area and rate were remarkably high" - providing some statistics here would be beneficial.
- The final sentence of the abstract doesn't feel particularly well supported: how specifically does this information help design conversion-free supply chains? Surely those could be developed without the information relayed by this manuscript? However, the manuscript certainly helps to understand potential trade-offs when designing supply chains, particularly concerning forest protection (and its unintended spillover effects) and potential biodiversity impacts.

Main text:

- There is some repetitive phrasing when discussing the "leakage" effects of zero-deforestation policies.
- The justification for the different land-cover datasets is helpful, though it would be useful to know if there are any common limitations.
- Before and in Figure 1, it is unclear what years you are using to detect conversion; please clarify that you compare 2000 with 2010 and 2010 with 2020 to assess conversion rates across decades.
- Figure 1b - The colour scale is hard to see on the map, but I recognise the challenge of relaying so much information in a single figure.
- Figure 1c - What criteria were used to define a hotspot country or region? This is mentioned throughout the text, but there is no clear definition. Is it a country where three of the land-cover maps agreed on conversion, or is there a threshold level of conversion? Without more information, it is hard to determine whether these are hotspots of conversion or just areas with high levels of remaining non-forested lands.
- On p.6, when saying that there is more non-forest than forest conversion, it would help if you gave the reader an indication of how much of the world is covered by non-forest vs forest (these data are shown in Fig S1).
- At the start of p.7, the authors provide the overall non-forest conversion, but not the overall forest conversion - including this would be helpful since it is one of the key results in the abstract.
- Fig 2a - the figures up to this point have shown results for forests for comparison - it would be more consistent to include that here as well.
- There are far fewer endemic species in grasslands and wetlands at risk from conversion compared to those at risk from forest conversion; this result is overlooked in favour of stating that many forest species at risk also rely on other habitats. It would be sensible to acknowledge this result and the importance of forest habitats; this does not undermine the conclusion

that non-forests are still valuable and should not be ignored.

Methods:

- The first paragraph is unclear. It suggests that conversion was calculated by "overlaying land-cover maps of the initial and final years under study". This reads as 2000 and 2020, but it was 2000 with 2010 and 2010 with 2020. Please clarify.
- The species analysis was only done with the land-cover dataset that accounts for cultivated pasture vs natural grassland, but does not account for tree plantations and is potentially more prone to misclassification of croplands. If repeating the analyses for all land-cover maps is not possible, please suggest how the land-cover map's limitations might influence the results.

Supplementary Material:

- Fig S5 is mentioned in the main text as a check for spurious changes; it would be helpful to include explanatory text on this assessment and an interpretation of the results in the Supplementary Material.

Code:

- It might assist readers if a comment is added in the code to explain why map.t1 and map.t2 are multiplied by different values.

(Remarks on code availability)

I have read through and tested parts of the code, which seems well-commented and robust (please see comment in the main review). However, I have not run the code in full so have not reproduced the results.

Version 1:

Reviewer comments:

Reviewer #1

(Remarks to the Author)

Interesting and relevant study on global hotspots of agricultural conversion, and the environmental impacts of conversion. By identifying these hotspots, focused attention can be placed on conservation measures. The authors did an extremely thorough job of addressing my previous reviewer comments, as well as the comments from two other reviewers.

(Remarks on code availability)

Reviewer #2

(Remarks to the Author)

Thank you for the detailed response to my review. You have answered my questions satisfactorily. Two additional comments:

"We would, however, expect the scale of non-forest conversion driven by tree crops to be rather small at the global level, as converting land that naturally carries short vegetation to tree crops or tree plantation is highly input intensive." I would challenge this. For a number of reasons - not least conservation efforts towards forests - grasslands have become less controversial places for land use conversion. Large areas of African savannah are identified for afforestation for instance. I don't think this makes a material difference to your analysis but perhaps a mention in the text.

"HILDA shows that the conversion from unmanaged grassland/shrubland to pasture/rangeland during 2000 2020 was almost twice the conversion from pasture/rangeland to cropland, indicating that our results might underestimate non-forest conversion driven by total agricultural activities." If this were a robust conclusion it would be very important; focusing attention on (from a conservation perspective) loss of grassland quality as being more significant than conversion. But I note your caution on these figures. You need to decide if the data are strong enough to comment on this

(Remarks on code availability)

Reviewer #4

(Remarks to the Author)

General comment.

Before starting my evaluation, I must clarify that this review follows two other evaluation processes initiated by other reviewers, which have improved the quality of the manuscript.

Using secondary information (three general global models of land covers, limits of protected areas, and range-polygons of

all the threatened amphibians, mammals, reptiles provided by IUCN), authors of this manuscript sought to 1) globally identify hotspots of conversion from non-forested natural areas to non-natural covers, 2) evaluate this landcover change in protected areas, and 3) estimate the possible effect of this landcover change on the conservation of threatened species. In general terms, I found that the methods proposed to develop the research objectives were well executed, which would allow for generating results consistent with used resources. However, my conclusion is that the manuscript should not be published in Nature but it can be published in another Springer journal that has lower requirements in relation to quality of the data and novelty. The details of this decision are set out below.

Clearly, conducting a precise and comprehensive analysis of how non-forested natural areas have been transformed by anthropogenic covers would be a major step forward the conservation of these areas on the planet. However, the main problem that I identified is the low confidence of the landcover maps used for the analysis. It is known that these three landcover global maps (GlobeLand30 25, Global Land Cover and Land Use Change -GLCLUC, GLC_FCS30D) do not have enough local accuracy in identifying non-tree natural covers, such as savannas, grasslands, and wetlands against non-natural grasslands, such as, non-natural pastures, crops and tree and shrub plantation. Consequently, the results shown in this work have inaccuracies that cannot be controlled by authors.

At the last part of the introduction, authors suggested that this study can help to generate conservation and planning strategies on non-forest natural vegetation (a point already proposed by other authors using the same used global maps of this work) and clarify that this work did not intend to precisely quantify conversion rates of non-forest natural vegetation. The issue is that we already have other similar analysis (as in this study) that proposed that we are at the point where it is necessary precisely quantify conversion rates of non-forest natural vegetation. The previous show that this work is not a novel analysis at level of Nature.

Following with the novelty point, authors added in the introduction information on global and regional estimates of conversion, carbon stock and biodiversity in grassland and wetlands, showing that we already have similar global analyses. Also, other authors have already published global analysis on grassland and wetland conversion; therefore, the presented analysis could not be considered as a total novel analysis, which is not negative, but then this would not meet Nature's novelty requirements. The next some other similar global analysis already published:

- Annual al 30-m maps of global grassland class and extent (2000–2022) based on spatiotemporal Machine Learning: <https://doi.org/10.1038/s41597-024-04139-6>
- Global wetland maps: <https://www2.cifor.org/global-wetlands/>
- The Global Lakes and Wetlands Database (GLWD) version 1 and 2 (it includes mapping and analysis between 1990-2020: <https://www.hydrosheds.org/products/glwd>).
- Global Wetland Watch: mapping and monitoring changes to wetland ecosystems: <https://unepdhi.org/global-wetland-watch-mapping-and-monitoring-changes-to-wetland-ecosystems/>

Based on the criteria established by Nature for its publications, my conclusion is that this work does not meet the standards of precision and novelty that a publication in Nature must have to. I realize we are at the point where we need the results of precise analyses focused on non-forested natural areas and not what this work does; this work reanalyzed general global maps where the classification error of non-forested areas is too high locally and/or cannot be controlled by the authors. Authors' conclusion is obvious: non-forested natural areas have decreased over time, this work would be enough 10 years ago, but we need more now. We now need more than obvious general trends. We need spatial details about how these trends change within country regions, which are the analyses that this paper doesn't show. My recommendation is that the author look for a lower impact factor journal to submit.

(Remarks on code availability)
Additional specific comments

Pag6. Why did authors provide a range for forest cover and non-forest natural land covers? (At a global scale, there was extensive and increasing non-forest natural land covers spanned 3.0 – 5.5 Gha in 2000 while forest areas covered 4.1 – 5.1 Gha). It means that the estimates of the three used maps are highly contrasting and would be very inaccurate. So again, we're at a point where the analysis presented is correct, but we need to use more accurate maps. Thus, we need these types of maps first.

Pag7. This result is not expected (The variations across the three datasets for non-forest conversion is much smaller than that for forest conversion). In terms of remote sensing, the classification of forests is easier than other vegetation (grassland, wetlands crops, etc.) because covering, photosynthetic activity, and structure are more stable in forest than crops and non-forest natural land covers.

Version 2:

Reviewer comments:

Reviewer #4

(Remarks to the Author)

As I concluded previously, all methods proposed to develop the research objectives were well executed, which allowed for generating results consistent with used resources. As well, it is valuable that authors have worked and improved all the minor comments that I suggested in my first revision. Also, I have evaluated carefully the responses of authors to my two main criticisms to the manuscript: 1) the three landcover maps used by authors have inaccuracies that authors cannot control and 2) manuscript does not meet the standards novelty that a publication in Nature must have to.

Regarding the first criticism, authors have provided all the evidence to show that these global land cover maps are the best models we have for assessing the loss of non-forested natural areas, and they emphasized that any remote sensing analysis (such as these three novel maps and their respective publications) is inherently flawed. The authors explain that they linked the three maps to increase the confidence of their analysis, and I agree with this. My conclusion regarding this first criticism is that, by combining the information from the three maps, the authors have some control over, improving the confidence of the maps, thus providing certainty to the analysis in this manuscript. Therefore, I concluded this point is resolved.

Regarding the second criticism, I emphasize that this manuscript is a reanalysis of three published models of land covers (where each one was novel analysis by the time its publication) and other published species distribution models. Therefore, I believe this manuscript does not constitute a novel analysis. However, I reviewed Nature's publication standards, and it is not clear whether a reanalysis of previously published models constitutes a novel analysis. I personally conclude that the analysis is not novel, consequently I suggest the article should be published in a lower-impact journal. However, I prefer that the decision be made by the editor in chief.

(Remarks on code availability)

I did in my first review.

Response to the comments

We sincerely appreciate the reviewers' time and effort in evaluating our work, as well as the thoughtful and constructive comments. We have carefully addressed each point and made substantial revisions accordingly. We believe these revisions have strengthened the manuscript and improved its clarity and reliability. We hope that our responses satisfactorily address all concerns.

Please find a detailed response to all reviewer comments (**in bold**) below, with crucial information highlighted in blue font.

REVIEWER COMMENTS

Reviewer #1 (Remarks to the Author):

Comment:

Interesting study on a topic that does not receive much attention. There are two major points I would like to see addressed.

The first is to explain why Canada is not identified as a primary hotspot. Canada is the second largest country, by land mass, in the world. Canada has vast prairie grasslands. In addition, and perhaps more importantly, three of the largest wetlands in the world are found in Canada: The Hudson Bay Lowland, The Mackenzie River basin, and the prairie potholes of North America (shared with USA). Based on these facts it seems to me impossible that Canada would not be identified. And because of that, I question the data.

Response:

We sincerely appreciate your thoughtful and insightful comment. You raise a very important point regarding Canada's vast grasslands and wetlands, and we fully understand how its absence as a primary conservation hotspot might be surprising. Your feedback prompted us to carefully validate our findings with additional literature to ensure the robustness of our results and to further explore potential explanations for this observation.

While Canada did not rank among the top 10 hotspot countries, it consistently appeared within the top 15 across all three datasets we examined. However, the presence of

extensive natural vegetation does not necessarily translate into higher conversion area.

Several key factors help explain this:

1. Comparatively Small and Declining Cropland Extent

According to the Food and Agriculture Organization (FAO) ¹, Canada's total cropland area is smaller than that of the most prominent conversion hotspot countries, over four times smaller than India and the United States and over three times smaller than China and Russia in 2020. Moreover, it has declined by nearly 10% between 2000 and 2012, and despite some recovery in subsequent years, the 2020 cropland area remained approximately 7% lower than in 2000. The Canadian Census of Agriculture also reports a long-term decline in total farm area across all provinces, with a national decrease of 8% over the past two decades (from 68 million hectares in 2001 to 62 million in 2021) ². Although localized cropland expansion may still occur in some pixels (i.e., gross increase), the overall net decline can reflect broader socioeconomic dynamics (e.g., rising farmland prices ³) to some extent, which are influencing the general trend of Canadian cropland extent and area.

2. Limited Overlap between Agricultural Suitability and Non-forest Natural Areas

Our datasets, along with additional land cover data from the European Space Agency ⁴, previous literature and government report ⁵⁻⁷ show that, cropland in Canada is primarily concentrated in the Prairies, not located in the areas of the major remaining non-forest ecosystems, as a combined result of various factors influencing agricultural suitability, such as climate, water availability, distance from traffic network and living areas. The localized distribution further minimizes the interface between cropland and adjacent natural areas, where agricultural expansion typically occurs ⁸⁻¹⁰, and increases the difficulty of accessing more remote natural areas. In contrast, many hotspot countries, such as Russia, the United States, and China, have much more dispersed cropland distribution, which partly indicate more spatial overlap between agricultural suitability and natural areas, as well as easier access to these areas.

3. Influence of Socioeconomic Factors

Natural land conversion is also influenced by socioeconomic factors, such as policy regulations and environmental awareness ^{11,12}. Canada has implemented a series of natural ecosystem conservation policies, such as The Federal Policy on Wetland Conservation ¹³, while countries with less conservation policies and weaker enforcement tend to experience more conversion. For instance, Canada has 4.8

times more non-forest natural lands covered by Protected Areas than Tanzania but experienced far less conversion within these Protected Areas. This can partly reflect the strength and effectiveness of its conservation actions and explain why Canada's total conversion of non-forest natural lands was lower than Tanzania, despite Canada has over 3 times more croplands and 4.5 times more non-forest natural lands.

Considering these well-substantiated findings, supported by cross-validation across multiple datasets and many additional literature and reports, we believe our findings are reliable. Nevertheless, we fully acknowledge the importance of this point and have incorporated further explanations into the discussion section to ensure clarity for readers:

“The spatial distribution of conversion hotspots is influenced by both geographic and socio-economic factors. Key geographic determinants include the extent and spatial distribution of agricultural and natural non-forest lands. Major conversion hotspots, such as Brazil, Russia, China, and the United States, also have the largest non-forest natural land area. However, Canada ranks only around 15th in non-forest conversion, despite ranking around 5th in non-forest natural land area. This is primarily due to Canada experiencing agricultural land contraction in recent decades^{37,48}. Additionally, agricultural areas in Canada are primarily concentrated in the Prairies due to agricultural suitability, not located in the areas of the major remaining non-forest ecosystems. The localized distribution further minimizes the interface between cropland and adjacent natural areas, where agricultural expansion typically occurs^{49,50}, and increases the difficulty of accessing more remote natural areas. Socio-economic factors, such as policy regulations and environmental concerns, also play an important role⁵¹. Countries with less conservation policies and weaker enforcement tend to experience more conversion. For instance, Canada has 4.8 times more non-forest natural lands covered by PAs than Tanzania but experienced far less conversion within these PAs. This can partly reflect its stronger conservation efforts and explain why its total non-forest conversion was lower than Tanzania, despite having over 3 times more croplands³⁷ and 4.5 times more non-forest natural lands.”

Comment:

The second is to account for climate change. There is no mention of climate change in this study. There are vast regions of non-forested tundra and permafrost affected by climate change. There is interest in expanding agriculture into these areas as they become warmer. It would be valuable and interesting to forecast

climate change effects on tundra and permafrost as they relate to agricultural expansion.

Response:

Thank you very much for your valuable suggestion. We completely agree with you on the importance of addressing the impact of climate change on tundra and permafrost, areas increasingly threatened by warming temperatures. Actually, we have also been concerned about this issue and have been exploring it step by step. For example, considering the primary focus on tropical deforestation in existing literature, our previous research examined the loss and drivers of boreal and temperate intact forest landscapes, which include significant tundra and permafrost regions and are increasingly vulnerable to human activities ¹⁴.

However, it is particularly challenging for us to forecast climate change effects on tundra and permafrost. To our best knowledge, the quantification requires integrated earth systems simulation models that account for complex interactions between physical, chemical, biological and human systems. While we recognize the rapid and exciting advances in earth system modelling over the past decades, thanks to the efforts of earth system scientists, we do not have the expertise or research experience in this field to conduct such analysis ourselves.

Nevertheless, your comment has prompted us to estimate agricultural expansion into tundra and permafrost areas in this study. Since GLCLUC dataset does not include separate land cover classes for tundra or permafrost, and due to the differing standards of the 3 datasets examined in distinguishing dense short vegetation, sparse short vegetation, and bare areas, we took the following approach: (1) We used a global map of ecoregions categorized into 14 terrestrial biomes ¹⁵, and focused on conversion within “Tundra” and “Rock and Ice” biomes in boreal regions; (2) We aggregated all terrestrial land covers that are not classified as forest, cropland and artificial areas. Our results show increasing but very small conversion area (less than 400 ha) in Tundra and Rock and Ice biomes in Canada, Iceland, and the United States.

Despite these challenges, we totally agree that this issue is of high importance, and we are also very interested to explore collaboration opportunities to forecast and better understand the influence of climate change. The challenges underscore the urgent need for more interdisciplinary earth system modelling studies supported by scientists from diverse scientific background. Therefore, in the discussion section, we have emphasized the significance of studying the climate change impact on tundra and permafrost:

“In addition to existing conversion hotspots, it is also essential to pay more attention to non-forested tundra and permafrost regions. These areas store vast amounts of carbon but are increasingly threatened by global warming, which causes thawing and makes them more accessible to human activity⁴². Accordingly, more interdisciplinary studies are needed to project the impacts of climate change on these vulnerable regions.”

Reviewer #2 (Remarks to the Author):

Comment:

Thank you for an interesting publication. There is definitely a need to focus on non-forest ecosystems and your general conclusions are in line with other studies of this nature. However, I have some comments about the scope of the manuscript.

In the introduction, it would be useful to have slightly more emphasis (an explanatory paragraph) on why conservation of non-forest lands is important. This is necessary here because the arguments for a focus on forests have generally related to their unique richness. Grassland is, for instance, believed to be sometimes a better ecosystem for carbon storage than forests in areas where forests burn regularly, even though per area carbon storage is less.

Response:

Thank you very much for your valuable suggestion! We have incorporated additional explanations in the first paragraph to further highlight the importance of non-forest ecosystems, which we believe can also strengthen the significance of our analyses. The revised paragraph was pasted below for your reference:

“Non-forest natural ecosystems, including natural grasslands and wetlands, are of great ecological significance globally. Grasslands store over one-third of global terrestrial carbon and have the potential to sequester 2.6 to 8.0 billion tons of carbon dioxide equivalents per year through soil organic carbon storage^{7,8}. They also hold over 33% of global biodiversity hotspots and contain habitats of global importance⁹. Due to their greater resilience to extreme heat-waves, drought and wildfire compared to forests, they are considered as more stable and reliable carbon sinks in areas where forests are climate-vulnerable¹⁰. As reported by the Convention on Wetlands and previous literature, wetlands harbor 40% of all known animal and plant species (including numerous endemic species)¹¹ and hold over 30% of the world’s soil organic carbon¹², despite covering only 6% of the global land surface.”

Comment:

I was looking for more discussion about how you are defining the three biomes considered and how the GlobalLand30 classification referred to has been interpreted in practice. “Wetland” is interpreted in very different ways around the world – the Convention on Wetlands includes coastal waters to 6 metres depth for instance, which I don’t think you are including here, so it would be worth defining this clearly in the main body of the paper. Furthermore, I’m not sure the sources you are quoting in the introduction are necessarily using the same definitions. If peatlands are included in wetlands my guess is that the figure for 20% global soil carbon is an underestimate. Similarly, do your estimates cover the whole land surface, in which case where does tundra sit – within wetlands or grasslands (some but not all definitions of rangelands include tundra because of reindeer herding and use of caribou)? I think a statement is needed of what you understand has and has not been included in the various definitions (including differences between the three data sets, which you have tried to do).

Response:

Thank you very much for this highly constructive comment! To enhance clarity, we have provided further explanations about how the three biomes are classified across the three databases, in Table 1 and the 5th paragraph in the results section. To maintain conciseness in the main text, we kept these explanations relatively brief; however, we have provided a detailed description of the classification schemes in the Supplementary Information Table S1.

Table 1 Land cover classifications in this study and corresponding classes in each dataset

This study	Cultivated land/Cropland	Grassland & Shrubland	Non-Forested Wetland	Forestland
GlobeLand30	Cultivated land (Exclude tree crops Include cultivated grassland)	Grassland and shrubland	Wetland	Forestland
GLCLUC	Cropland (Exclude tree crops Exclude cultivated grassland)	Short vegetation and tree cover < 5m in Terra Firma class	Short vegetation and tree cover < 5m in Wetland class and open surface water present 20- 79% of the year	Tree cover ≥ 5m, incl. forested wetlands
GLC_FCS30D	Cropland (Include tree crops Exclude cultivated grassland)	Grassland and shrubland	Marsh, salt marsh, flooded flat, tidal flat, saline	Forestland, swamp, mangrove

Specifically, regarding your comment on wetlands, wetlands are generally defined as lands transitional between terrestrial and aquatic systems and all the three datasets exclude coastal waters, but only GLCLUC provides a clear quantitative identification criterion. Due to our focus on non-forest ecosystems and the fact that many existing studies on deforestation focus on tree cover loss — often categorizing forested wetland loss as deforestation^{16,17}, we have revised the classification scheme for wetlands and forestlands in our study to align with these literature. GLCLUC and GLC_FCS30D contain several wetland subcategories, enabling classification of forested wetlands under forestland. Following FAO's definition that forest should have trees higher than 5 meters, tree cover $\geq 5\text{m}$ in GLCLUC's wetland class was classified as forestland. Similarly, GLC_FCS30D's swamp and mangrove (both covered with woody vegetation) were classified as forestland, which provides a conservative estimate of non-forested wetlands as some woody land covers are not forests. In contrast, this distinction was not possible in GlobeLand30. Considering that non-forested wetland area was 1.5 times larger than forested wetland area in 2000 (according to GLC_FCS30D), the entire wetland class in Globeland30 was excluded from the forestland class. Notably, total wetland conversion in GlobeLand30 was smaller than non-forested wetland conversion in the other two datasets, indicating differences in their classification standards. Generally, this revision has little impact on our findings, as the total area of wetlands is much smaller than other natural land covers. We have further explained the definitional differences in the manuscript.

Regarding the sources cited in the Introduction, we acknowledge that the definition of wetlands was not sufficiently clear, and that peatlands may not have been included, potentially leading to an underestimation of global soil carbon storage in wetlands. To ensure greater accuracy, we have carefully reviewed the latest literature and reports, and revised the relevant content of the introduction. We found that, inland freshwater wetlands and peatlands alone hold over 30% of the world's soil organic carbon. However, wetlands may be defined differently in different studies, for example, all the three datasets used in this study do not provide an individual class for peatland. In order to avoid misunderstandings, we have clarified the data sources so that readers can recognize that the statistics correspond to the wetlands as defined in those data sources:

“As reported by the Convention on Wetlands and previous literature, wetlands harbor 40% of all known animal and plant species (including numerous endemic species)¹¹ and over 30% of the world's soil organic carbon¹², despite covering only 6% of the global land surface.”

Regarding your comment on tundra, the three datasets have differing classification standards. GlobeLand30 and GLC_FCS30D include a separate Tundra class, whereas GLCLUC does not. GlobeLand30 defines it as lands covered by lichen, moss, hardy perennial herbs, and shrubs in cold and high mountain areas, including shrub tundra, grass tundra, wet tundra, alpine tundra, and barren tundra. GLC_FCS30D defines it as lichen and moss, without further explanation. However, in GLCLUC, vegetation in tundra areas is allocated to tree cover and short vegetation. We have included an explanatory note on this issue in the results section.

Besides explanations added in the results section, we have also highlighted the urgent need of a clear and consistent land cover classification scheme in the discussion section.

Comment:

While the general conclusions seem logical, i.e., the top-ranking countries for conversion, I'm confused by some of the figures, particularly related to protected areas. I assume from the caption on figure 2 that analysis is by country but find the reference to outlier data surprising. Are you saying that some countries have lost more than 90% of their non-forest ecosystems inside protected areas over the last 20 years? If this is really the case this is dramatic news and the countries need to be identified. But I would be looking for a lot of verification with country experts before publishing this.

Response:

We really appreciate this insightful comment! The phenomenon is partly due to the uncertainties of the three datasets, while the main reason is our unclear definition of conversion rate and limitations of our previous calculation method. To be more specific, if there were not newly established PAs since 2000, then the share of conversion within PAs between 2000-2020 can be calculated as:

$$\frac{\text{Conversion within } PA_{i,2000} \text{ in } 2000 - 2010}{\text{Area of } PA_{i,2000}} + \frac{\text{Conversion within } PA_{i,2000} \text{ in } 2010 - 2020}{\text{Area of } PA_{i,2000}} \quad \text{Eq.(1)}$$

where $PA_{i,2000}$ represents PAs established before 2000 and covered by the land cover type i in 2000.

For 2010 – 2020, we intended to include conversion within newly built PAs during 2000-2010, the conversion rate was consequently revised as:

$$\frac{\text{Conversion within } PA_{i,2000} \text{ in } 2000 - 2010}{\text{Area of } PA_{i,2000}} + \frac{\text{Conversion within } PA_{i,2000} \text{ and } PA_{i,\text{newly built } 2000-2010} \text{ in } 2010 - 2020}{\text{Area of } PA_{i,2000} + \text{Area of } PA_{i,\text{newly built } 2000-2010}} \quad \text{Eq. (2)}$$

where $PA_{i,\text{newly built } 2000-2010}$ represents PAs newly built during 2000 and 2010.

Therefore, the conversion rates calculated based on our previous method can partly reflect the severity of conversion within PAs but does not exactly mean share of PAs that were converted, which might cause some misunderstanding. Please see the example below, which can misguide readers overestimate the share of converted PAs :

Total area of PAs in 2000	100
Conversion 2000 - 2010 within PA2000	90
Conversion rate 2000-2010	90%
Total area of PAs in 2010	10000
Conversion 2010 - 2020 within PA2010	0
Conversion rate 2000-2010	0%
Total conversion rate according to our previous method	90%+0%

To address this issue, three revisions have been made:

- (1) We have modified the calculation method: since we focus on the share converted during the entire 2000-2020 period, only PAs that were established before 2000 were considered (i.e. **Eq. (1)**), which is more consistent with instinctive cognition. This has been clarified in the methods section.
- (2) We compared results across all the three datasets, considering that the high rates observed in one dataset may be due to its misclassifications.
- (3) We have explored reasons for the high conversion rates observed.

Please see **Fig. 1** below for the updated results based on the revised method. The conversion rates are much lower than previously estimated: for 96% (GLC_FCS30D) to 99% (GLCLUC) of the approximately 180 countries having protected areas in 2000, conversion rates remained below 60%. In countries exhibiting high conversion rates for a specific PA category, the total PA area in that category is typically small. The high conversion rates may result from limited enforcement of agricultural restrictions in

those regions (e.g., in Cambodia¹⁸), or from dataset misclassifications (as even misclassifications of a few pixels within these PAs can lead to substantial differences in conversion rates), which may explain why no country with a conversion rate exceeding 60% appeared consistently in at least two datasets. However, in general, GLCLUC and GlobeLand30 datasets show similar patterns at a global scale and all the three datasets consistently show that conversion rates for non-forest PAs were higher than forest PAs.

Fig. 1 Conversion within Protected Areas (PAs) per country

a. The proportion of conversion within each IUCN PA category relative to the total conversion within all PA categories; **b.** The percentage of conversion inside PAs (i.e., conversion inside PAs/total PA area) by IUCN PA category and the percentage outside PAs (i.e., conversion outside PAs/total non-PA area, marked as “No” in the figure). Each column corresponds to a combination of a specific IUCN PA

category and the corresponding land cover type covering PA areas as of 2000, summarizing conversion rates across countries. The upper and lower boundaries of each box and the horizontal lines inside respectively represent the upper quartile, lower quartile, and median conversion rates. The upper and lower boundaries of each whisker represent the minimum and maximum values within 1.5 times the interquartile range. The dots represent outliers where conversion rates are higher or lower than 1.5 times the interquartile range. For display purposes, the vertical axes are capped to 60%, which include 96% (GLC_FCS30D) to 99% (GLCLUC) of about 180 countries having PAs in 2000.

Comment:

Similarly, while the caveats in the discussion section were very useful, they also throw question on some of the results. If existing methodologies cannot differentiate pasture from natural grassland (and this includes being unable to distinguish pasture resown with non-native species), or tree crops from natural forest, then what impact does that have on the results? I see that one of the three datasets used claims to include cultivated pasture amongst “cultivated land/cropland) but how is this done? Do you have opinions about the accuracy?

Response:

1. Regarding the impact of not differentiating pasture / tree crops from natural vegetation

It is difficult to determine whether our estimates are higher or lower than the total conversion of natural land covers due to all agricultural activities. In the case of cultivated pastures not being distinguished from natural lands, our estimates may exclude conversions from natural grasslands to cultivated pastures (underestimating total agriculture-induced conversion of natural lands) while including conversions from cultivated pastures to croplands (overestimating it). The same issue applies to tree crops and natural forests. We would, however, expect the scale of non-forest conversion driven by tree crops to be rather small at the global level, as converting land that naturally carries short vegetation to tree crops or tree plantation is highly input intensive.

To preliminarily assess these impacts, we compared three datasets—one of which includes tree crops in cropland and another includes cultivated pasture. However, due to many other differences in land cover classifications, it remains unclear what are the main drivers of the discrepancies in results. For example, GLC_FCS30D shows more forest conversion to cropland than the other two datasets. This could be due to GLC_FCS30’s lower tree canopy threshold for defining forests (identifying more converted tree cover as deforestation) or its differentiation between tree crops and natural forests.

For further validation, we have carried out additional comparison with existing work in this field, based on the HILDA dataset produced in that work ¹⁹ (Supplementary Information Fig. S5). It differentiates cultivated pasture from both croplands and unmanaged grasslands, allowing for valuable insights into the impact of not making this distinction on conversion estimates. HILDA shows that the conversion from unmanaged grassland/shrubland to pasture/rangeland during 2000-2020 was almost twice the conversion from pasture/rangeland to cropland, indicating that our results might underestimate non-forest conversion driven by total agricultural activities. However, due to the lack of high-resolution pasture distribution data, HILDA identifies pasture based on FAO's Gridded Livestock World v3 2010, which has a coarse resolution of 5 arc minutes (~9.28 km at the equator) and was only available for a single year, rather than directly using annual satellite imagery. Moreover, HILDA was compiled at 1 km resolution using majority cell values, which can lead to misestimation as we do not have access to more detailed land cover fraction data. For instance, a 1 km² pixel marked as forest-to-cropland conversion may contain other land use changes within it, but we have to assume all the 1km² were converted from forest to cropland. Therefore, the results should be also interpreted with caution due to inherent uncertainties. Further analyses are still required to quantify such impacts.

In order to address this issue, we have tried our best to emphasize that the readers should pay particular attention to the land cover definitions and scopes of each dataset when interpreting results. We have also added further explanations of these definitions in the results section (see response to prior comments), as well as the importance of a more accurate and better-defined land cover products in the discussion section.

2. Regarding GlobeLand30's identification of cultivated pastures and its accuracy
GlobeLand30 does not explicitly classify cultivated pastures but identifies lands with evidence of human cultivation. It employs a three-step method integrating phenology of crops and cultivated grasses (e.g., planting, growth, maturity, and harvest), regular cultivated land distribution patterns (e.g., circular or rectangular fields), and expert validation. First, a pixel-based method extracts all pixels displaying phenology traits of cultivated crops and grasses. To our understanding, this step differentiates cultivated pastures from natural grasslands based on agricultural patterns like rotation and homogenization. Second, an object-based classification and image segmentation method groups pixels with uniform spectral responses into structural objects, each potentially representing the same land use

and land cover type within its boundary; Third, the segmented objects were overlaid onto potential cultivated land pixels. Only those with regular man-made patterns (e.g., circles, rectangles) and at least 70% potential cultivated land pixels were classified as cultivated land through visual interpretation. Finally, a group of experts verifies the results.

For accuracy assessment, 159,874 pixel samples from the 2010 baseline year were compared to Landsat TM imagery. The overall accuracy for cultivated land is 82.76%. There are also some studies evaluating accuracy for cultivated land at national or regional scales, such as Kyiv Oblast of Ukraine , Shannxi Province of China, the whole of China and East Africa, all of which report user's accuracies between 80% - 90% ²⁰⁻²³.

Comment:

Finally, I had to read figure 2 several times to make sense of the figures, whereas figure S4 was immediately clear. I suggest including the latter in the main body of the paper.

Response:

Thank you for your feedback. We have carefully designed our figures to present the information as clearly as possible and explored alternative formats but found no better representations. To improve clarity, we have added more detailed descriptions in the figure captions to aid interpretation. Additionally, we have incorporated previous Figure S4 into Figure 1 in the main body of the paper and revised previous Fig. 1b (now Fig. 1a) to enhance clarity. We appreciate your suggestion, as these revisions improve the overall readability of the figures.

Reviewer #3 (Remarks to the Author):

This analysis was interesting, with great care taken to choose the land cover datasets and explain their benefits and limitations. The manuscript outlines the importance of non-forest systems for biodiversity and highlights the vulnerability of these systems (and the species that depend upon them) to conversion into cultivated lands (e.g., cropland). The results emphasize that the conversion of these lands has far outstripped that of forested lands over the last two decades, with rates of conversion increasing over time.

General comments:

Generally, I felt that the work was robust with a sound methodology, thanks to the use of multiple land cover datasets to find general patterns. However, I felt the link to policy - in particular, the idea of designing conversion-free supply chains - lacks some details and logic, leaving it feeling a little forced. I was surprised that Winkler et al. (2021) (<https://www.nature.com/articles/s41467-021-22702-2>) was not referenced; it would be helpful to compare the methods and results with this previous analysis, which also looked at land-use conversion rates over time. In some places, details on the results or methods are lacking, making it difficult to assess the overall importance of some of the take-home messages, but these issues could be easily resolved. I also would have appreciated a more detailed assessment of why the spatial patterns in conversion might be observed. The authors have made an effort to clearly comment their code, making it much easier to review.

Response:

Thank you very much for these both constructive and concrete suggestions! We have made major revisions according to these comments point by point.

1. Regarding conversion-free supply chain policies

We acknowledge that the primary focus of this paper is to identify the conversion hotspots, which provide direct evidence-based insights for prioritizing future conservation areas. However, our results also serve as a starting point for helping design conversion-free supply chain policies for key supply chain actors, including companies and importer governments, which have been found to driven natural ecosystem conversion both directly and indirectly through their demand for conversion-risk agricultural commodities and their sourcing or trade of these commodities from conversion hotspot countries. Extensive research has demonstrated that consumption and trade have contributed to natural ecosystem conversion globally ^{24,25}, and many corporates and governments have implemented supply chain interventions to mitigate these impacts, but historically little attention has been given to non-forest ecosystems ²⁶. While this study does not attribute non-forest conversion to specific commodities or supply chains, most of the countries identified as conversion hotspots (e.g., Brazil, the US, Argentina) are major producers and exporters of known conversion-risk agricultural commodities ^{24,27}. This already suggests that companies and countries sourcing from these regions may be contributing to consequent biodiversity loss and carbon emissions. Therefore, they should adopt or extend their existing supply chain strategies to mitigate impacts across all ecosystems, not just forests.

In order to clarify this point, we have provided additional explanations in the discussion section to further elaborate on the implications of our findings for conversion-free supply chain policies and to highlight the need for future studies to link non-forest conversion to specific commodities driving this process. We sincerely appreciate this valuable comment.

2. Regarding comparison with existing literature (Winkler et al., 2021)

Thank you very much for this comment. We have added additional comparative analyses. The HILDA dataset (Winkler et al.¹⁹) differentiates cultivated pasture from croplands and unmanaged grasslands, allowing for valuable insights into the impact of not making this distinction on conversion estimates. Since Winkler et al. did not directly provide non-forest conversion estimates, we conducted additional analyses using the HILDA transition map, where each pixel is classified based on transitions between land cover categories. Specifically, we compared our estimates of grassland/shrubland and wetland conversion to cultivated lands/croplands with HILDA's conversion of unmanaged grass/shrubland (including wetlands) to cropland and pasture/rangeland.

It is important to note that, due to the lack of high-resolution pasture distribution data (largely stemming from technological limitations), HILDA identifies pasture based on FAO's Gridded Livestock World v3 2010, which has a relatively coarse resolution of 5 arc minutes (~9.28 km at the equator) and was only available for a single year. Therefore, the results should be also interpreted with caution due to inherent uncertainties.

Our findings align with HILDA in showing more extensive non-forest than forest conversion, though HILDA indicates a decreasing trend, as shown in **Fig. 2** below. Besides the treatment of pasture, several other factors contribute to this discrepancy, for example: (1) **Resolution differences**. The datasets in our study have a 30 m resolution, while HILDA was compiled at 1 km resolution using majority cell values, which can lead to misestimation as we do not have access to more detailed land cover fraction data. For instance, a 1 km² pixel marked as forest-to-cropland conversion may contain other land use changes within it, but we have to assume all the 1km² were converted from forest to cropland. (2) **Treatment of back-and-forth conversion**. Our estimates from the three datasets compared land cover 2000 with 2010 and compared 2010 with 2020 (because GlobeLand30 lacks annual data), therefore excluding conversion to cultivated lands/croplands that were later abandoned. In contrast, our estimates based on HILDA were obtained by summing its annual transition data, capturing these back-and-forth conversions, which can

affect comparability. This approach was chosen because comparing land cover changes over decadal periods at a coarse resolution may introduce greater uncertainties, as each pixel can encompass multiple land cover types. (3) **Land cover definitions.** Variations in classification criteria between datasets can also lead to differences in reported conversions, which has been elaborated in our manuscript. (4) **Uncertainties of each land cover dataset.** All the datasets rely on different data sources and methodologies to compile land cover maps, each subject to inherent inaccuracies and uncertainties.

We have added this comparison in the discussion section and included a figure with explanatory notes (Supplementary Information Fig. S5). Thank you again for your valuable suggestion!

Fig. 2 Comparisons of global conversion across four datasets

3. Regarding detailed assessments of the reasons for observed spatial patterns

In order to explain the spatial patterns of conversion, we have reviewed many additional literature and carried out further analyses. The key findings have been summarized in the discussion section and are pasted below for your reference:

“The spatial distribution of conversion hotspots is influenced by both geographic and socio-economic factors. Key geographic determinants include the extent and spatial distribution of agricultural and natural non-forest lands. Major conversion hotspots, such as Brazil, Russia, China, and the United States, also have the largest non-forest natural land area. However, Canada ranks only around 15th in non-forest conversion, despite ranking around 5th in non-forest natural land area. This is primarily due to Canada experiencing agricultural land contraction in recent decades^{37,48}. Additionally, agricultural areas in Canada are primarily concentrated in the Prairies due to agricultural suitability, not located in the areas of the major remaining non-forest ecosystems. The localized distribution further

minimizes the interface between cropland and adjacent natural areas, where agricultural expansion typically occurs^{49,50}, and increases the difficulty of accessing more remote natural areas. Socio-economic factors, such as policy regulations and environmental concerns, also play an important role⁵¹. Countries with less conservation policies and weaker enforcement tend to experience more conversion. For instance, Canada has 4.8 times more non-forest natural lands covered by PAs than Tanzania but experienced far less conversion within these PAs. This can partly reflect its stronger conservation efforts and explain why its total non-forest conversion was lower than Tanzania, despite having over 3 times more croplands³⁷ and 4.5 times more non-forest natural lands.”

4. Regarding other details on results and methods, we have made corresponding revisions according to your specific comments below. Please see our response for more details.

Specific comments:

Abstract:

Comment:

- This was well written and interesting, however, it is lacking in some key statistics to support the take-home messages. For instance "protected non-forest ecosystems cover substantially less area than protected forests, while their conversion area and rate were remarkably high" - providing some statistics here would be beneficial.

Response:

Thank you very much for this comment. We totally agree that adding key statistics in the abstract can help convey a clearer message to the readers. However, as explained in the main body of the manuscript, the statistics are highly dependent on corresponding land cover classification schemes used, and it is challenging to provide precise definitions within such a brief abstract. In order to avoid potential misunderstandings, we believe it would be better to focus on the common trends consistently observed across the three datasets.

Comment:

- The final sentence of the abstract doesn't feel particularly well supported: how specifically does this information help design conversion-free supply chains? Surely those could be developed without the information relayed by this manuscript? However, the manuscript certainly helps to understand potential

trade-offs when designing supply chains, particularly concerning forest protection (and its unintended spillover effects) and potential biodiversity impacts.

Response:

Thanks for the comment! We have replaced “supply chain policies” with “sustainable land-use policies” to encompass a broader range of interventions, because both companies and policymakers can implement measures beyond supply chain policies that still benefit non-forest ecosystems. However, we also believe our results have implications for supply chain policies, which we have discussed in detail in response to your general comment above. Please kindly refer to that section for further clarification.

Main text:

Comment:

- There is some repetitive phrasing when discussing the "leakage" effects of zero-deforestation policies.

Response:

Thanks for the concrete comment, we have revised the phrasing to avoid repeat:

“Previous sustainable land-use policies have typically targeted forests, but such a delimitation risks causing unintended leakage to non-forest ecosystems, compromising the environmental benefits of zero-deforestation policies^{14,15}. It is reported that Brazilian forests have been protected at the expense of native grasslands¹⁶. Similarly, zero-deforestation policies have also inadvertently increased the risk of oil palm expansion into grasslands in Southeast Asia¹⁷.”

Comment:

- The justification for the different land-cover datasets is helpful, though it would be useful to know if there are any common limitations.

Response:

Thank you for this suggestion. To our knowledge, all land cover datasets, including but not limited to the three examined in this study, are subject to inherent methodological limitations and potential misclassifications due to technological constraints. To reveal potential influence, we have summarized their user’s and producer’s accuracies and provided an additional explanatory note. Please refer to Supplementary Information Table S1(d) for further details. Additionally, we have also expanded our discussion in

Editorial Note: Administrative boundaries were modified from GADM version 4.1 in Fig. 3a.

the fifth paragraph of the Results section to better clarify limitations of the three datasets. Please see the manuscript for the detailed revisions.

Comment:

- Before and in Figure 1, it is unclear what years you are using to detect conversion; please clarify that you compare 2000 with 2010 and 2010 with 2020 to assess conversion rates across decades.

Response:

We have clarified this in the first paragraph in the results section: Conversion is assessed by comparing land cover in 2000 and 2010 for the period 2000–2010, and in 2010 and 2020 for the period 2010–2020.

Comment:

- Figure 1b - The color scale is hard to see on the map, but I recognize the challenge of relaying so much information in a single figure.

Response:

Thank you for this comment. To enhance clarity, we have redrawn previous Fig. 1b (now Fig. 1a in the revised figure) and included additional conversion distribution maps (Fig. S1 in the Supplementary Information), in order to illustrate both consistencies and inconsistencies across the three datasets. All the figures are also pasted below for your reference.

a

Editorial Note: Administrative boundaries were modified from GADM version 4.1 in Fig. 3b and c.

Fig. 3 Global distribution of conversion hotspots in 2000-2020

Values (mean from the 3 datasets) are presented as percentage of 30m by 30m pixels in a 10 km by 10 km grid cell for which such a conversion has been identified, for **a.** pixels where conversion is detected in **at least one** of the three datasets; **b.** Pixels where conversion is detected in **only one** of the three datasets; **c.** Pixels where conversion is detected in **two** of the three datasets. Only pixels within the top 90% of global conversion are colored.

Comment:

- **Figure 1c** - What criteria were used to define a hotspot country or region? This is mentioned throughout the text, but there is no clear definition. Is it a country where three of the land-cover maps agreed on conversion, or is there a threshold level of conversion? Without more information, it is hard to determine whether

these are hotspots of conversion or just areas with high levels of remaining non-forested lands.

Response:

Hotspot countries are those with highest average conversion areas across the three datasets, identified in at least two of them. Thanks for the comment. We have clarified this in the third paragraph of the results section.

Comment:

- On p.6, when saying that there is more non-forest than forest conversion, it would help if you gave the reader an indication of how much of the world is covered by non-forest vs forest (these data are shown in Fig S1).

Response:

Thanks for this comment, we have added the estimates to the second paragraph of the results section:

“At a global scale, non-forest natural land covers spanned 3.0 – 5.5 Gha in 2000 while forest areas covered 4.1 – 5.1 Gha.”

Comment:

- At the start of p.7, the authors provide the overall non-forest conversion, but not the overall forest conversion - including this would be helpful since it is one of the key results in the abstract.

Response:

We have also added this result to the second paragraph of the results section:

“Overall, non-forest conversion amounted to 173 – 243 Mha during 2000-2020, compared to 18 – 173 Mha of forest conversion.”

Comment:

- Fig 2a - the figures up to this point have shown results for forests for comparison - it would be more consistent to include that here as well.

Response:

We have included results for forests to ensure consistency. Please see Fig. 2 in the main body of the manuscript.

Comment:

- There are far fewer endemic species in grasslands and wetlands at risk from conversion compared to those at risk from forest conversion; this result is overlooked in favour of stating that many forest species at risk also rely on other habitats. It would be sensible to acknowledge this result and the importance of forest habitats; this does not undermine the conclusion that non-forests are still valuable and should not be ignored.

Response:

Thank you for pointing this out! We have revised related description:

“More threatened species endemic to forests were potentially affected than those endemic to non-forest ecosystems (2,310 vs. 625). Nevertheless, over 55% of the affected species are not endemic to forests and rely on non-forest ecosystems for their habitats. Notably, non-forested wetlands, despite their small land area, are home to over 1,500 threatened species, whether they are endemic or not.”

Method

Comment:

- The first paragraph is unclear. It suggests that conversion was calculated by "overlaying land-cover maps of the initial and final years under study". This reads as 2000 and 2020, but it was 2000 with 2010 and 2010 with 2020. Please clarify.

Response:

We have clarified that as follows:

“Specific areas of conversion were identified by overlaying land cover maps for the initial and final years of each period: the 2000 and 2010 maps for conversion between 2000 and 2010, and the 2010 and 2020 maps for conversion between 2010 and 2020.”

Comment:

- The species analysis was only done with the land-cover dataset that accounts for cultivated pasture vs natural grassland, but does not account for tree plantations

and is potentially more prone to misclassification of croplands. If repeating the analyses for all land-cover maps is not possible, please suggest how the land-cover map's limitations might influence the results.

Response:

Thank you for this valuable suggestion! We have added additional conversion distribution maps to illustrate inconsistencies between different datasets (Please see Fig. 3 in our reply to your comment above). These maps indicate that the impact of not considering tree crops is likely minimal:

(1) Since GLC_FSC30D includes tree crops in cropland, the potential impact of not considering them can be estimated by focusing on areas where GLC_FSC30D shows conversion but GlobeLand30 (what we used for species analyses) does not. It can be observed that such areas are relatively few and are mainly concentrated in Australia, India and Kazakhstan.

(2) Moreover, species habitats usually extend over large regions beyond individual pixels where differences between datasets occur. In these cases, we found that nearby areas were consistently identified as converted across all three datasets. This suggests that species whose habitats overlap with these pixels are likely already accounted for in the analysis.

In response to your suggestion, we have incorporated these analyses and explanations in the results section, further strengthening the reliability of our conclusions. We appreciate your insightful feedback!

Supplementary Material:

Comment:

- Fig S5 is mentioned in the main text as a check for spurious changes; it would be helpful to include explanatory text on this assessment and an interpretation of the results in the Supplementary Material.

Response:

Thanks for this advice. We have provided explanatory notes in the Supplementary Information:

“Misclassifications due to cropland fallow can occur when croplands are temporarily set aside for productivity recovery and misclassified as natural land cover. If cultivation

later resumes in these areas while other croplands are set aside, this can be mistakenly recorded as conversion of natural land covers to croplands. As a result, transitions from croplands to non-forest land may reflect both real changes, such as cropland abandonment and ecological restoration, and spurious changes due to misclassification. However, distinguishing between these factors remains challenging, limiting the ability to accurately quantify spurious changes. Some cropland identification techniques help mitigate these issues. For example, GlobeLand30 identifies cultivated lands by integrating both cultivated crop / grass phenology (e.g., rotation and homogenization) and regular cropland distribution patterns (e.g., circular or rectangular fields), which can help classify fallow croplands, as they typically follow these recognizable patterns.”

Comment:

Code:

- It might assist readers if a comment is added in the code to explain why map.t1 and map.t2 are multiplied by different values.

Code availability:

- I have read through and tested parts of the code, which seems well-commented and robust (please see comment in the main review). However, I have not run the code in full so have not reproduced the results.

Response:

Thank you very much for this comment. We have explained why map.t1 and map.t2 are multiplied by different values:

*“Map.t1 and Map.t2 are multiplied by different values and then added to Map.t3, in order to combine different land cover codes from layers for three years into a single code in a single layer, simplifying further analysis. For example, if a pixel is classified as grassland (code 30) in 2000 and as cropland (code 10) in both 2010 and 2020, its new code would be calculated as: $30*10000+10*100+10=301010$. This encoding allows the land cover classification for each year to be easily extracted later.”*

Reference:

1. FAOSTAT. Land, Inputs and Sustainability. (2025).
2. Statistics Canada, Government of Canada. Land use, Census of Agriculture historical data. <https://www150.statcan.gc.ca/t1/tbl1/en/tv.action?pid=3210015301> (2022).
3. Brockman, C. What's happening to Canada's farmland? *CBC News* (2023).
4. ESA. Land Cover CCI Product User Guide Version 2. Tech. Rep. maps.elie.ucl.ac.be/CCI/viewer/download/ESACCI-LC-Ph2-PUGv2_2.0.pdf (2017).
5. Potapov, P. *et al.* Global maps of cropland extent and change show accelerated cropland expansion in the twenty-first century. *Nat Food* **3**, 19–28 (2022).
6. Amani, M. *et al.* Application of Google Earth Engine Cloud Computing Platform, Sentinel Imagery, and Neural Networks for Crop Mapping in Canada. *Remote Sensing* **12**, 3561 (2020).
7. Government of Canada. Overview of Canada's agriculture and agri-food sector. (2024).
8. Graesser, J., Aide, T. M., Grau, H. R. & Ramankutty, N. Cropland/pastureland dynamics and the slowdown of deforestation in Latin America. *Environmental Research Letters* **10**, 034017 (2015).
9. Morton, D. C. *et al.* Cropland expansion changes deforestation dynamics in the southern Brazilian Amazon. *Proceedings of the National Academy of Sciences* **103**, 14637–14641 (2006).
10. Schielein, J. & Börner, J. Recent transformations of land-use and land-cover dynamics across different deforestation frontiers in the Brazilian Amazon. *Land Use Policy* **76**, 81–94 (2018).
11. Meyfroidt, P. *et al.* Middle-range theories of land system change. *Global Environmental Change* **53**, 52–67 (2018).
12. Rudel, T. K. *et al.* Forest transitions: towards a global understanding of land use change. *Global Environmental Change* **15**, 23–31 (2005).
13. Government of Canada. The Federal Policy on Wetland Conservation. <https://www.canada.ca/en/environment-climate-change/services/wildlife-habitat/federal-policy-on-wetland-conservation.html> (2024).
14. Kan, S. *et al.* Risk of intact forest landscape loss goes beyond global agricultural supply chains. *One Earth* **6**, 55–65 (2023).
15. Dinerstein, E. *et al.* An Ecoregion-Based Approach to Protecting Half the Terrestrial Realm. *BioScience* **67**, 534–545 (2017).
16. Curtis, P. G., Slay, C. M., Harris, N. L., Tyukavina, A. & Hansen, M. C. Classifying drivers of global forest loss. *Science* **361**, 1108–1111 (2018).
17. Hansen, M. C. *et al.* High-Resolution Global Maps of 21st-Century Forest Cover Change. *Science* **342**, 850–853 (2013).
18. Nuttall, M. *et al.* Protected area downgrading, downsizing, and degazettement in Cambodia: Enabling conditions and opportunities for intervention. *Conservation Science and Practice* **5**, e12912 (2023).
19. Winkler, K., Fuchs, R., Rounsevell, M. & Herold, M. Global land use changes are four times greater than previously estimated. *Nature Communications* **12**, 2501 (2021).

20. Yang, Y., Xiao, P., Feng, X. & Li, H. Accuracy assessment of seven global land cover datasets over China. *ISPRS Journal of Photogrammetry and Remote Sensing* **125**, 156–173 (2017).
21. Jacobson, A. *et al.* A novel approach to mapping land conversion using Google Earth with an application to East Africa. *Environmental Modelling & Software* **72**, 1–9 (2015).
22. Kussul, N., Shelestov, A., Basarab, R., Skakun, S. & Lavreniuk, M. Geospatial intelligence and data fusion techniques for sustainable development problems.
23. Chen, X. *et al.* Assessment of the cropland classifications in four global land cover datasets: A case study of Shaanxi Province, China. *Journal of Integrative Agriculture* **16**, 298–311 (2017).
24. Pendrill, F., Persson, U. M., Godar, J. & Kastner, T. Deforestation displaced: trade in forest-risk commodities and the prospects for a global forest transition. *Environ. Res. Lett.* **14**, 055003 (2019).
25. Hoang, N. T. & Kanemoto, K. Mapping the deforestation footprint of nations reveals growing threat to tropical forests. *Nature Ecology & Evolution* (2021) doi:10.1038/s41559-021-01417-z.
26. Lambin, E. F. & Furumo, P. R. Deforestation-Free Commodity Supply Chains: Myth or Reality? *Annu. Rev. Environ. Resour.* **48**, null (2023).
27. FAOSTAT. Production and Trade Statistics. <https://www.fao.org/faostat/en/#data> (2024).

Response to the comments

Part I - Cover letter to the reviewers

We sincerely appreciate the reviewers' valuable comments to improve our manuscript, which we have now resubmitted to the journal of *Nature Communications*.

We are especially grateful for the Reviewer #1's and Reviewer #2's satisfaction of our revision, noting that we had done "*an extremely thorough job of addressing my previous reviewer comments, as well as the comments from two other reviewers*" (Reviewer #1) and "*answered my questions satisfactorily*" (Reviewer #2). While Reviewer #3 was unavailable for the re-review, their original comments are truly encouraging, stating that "*This analysis was interesting, with great care taken to choose the land cover datasets and explain their benefits and limitations*" and "*I felt that the work was robust with a sound methodology*".

Reviewer #1 raised no further concerns, and while Reviewer #2 raised two additional minor questions, they also state that "*I don't think this makes a material difference to your analysis*". We have carefully addressed these points in the revised manuscript.

However, we were surprised and concerned by the comments from the newly invited Reviewer #4, especially as these comments contrasted with the consistently positive evaluations from all the three original reviewers, who have acknowledged the significance of our study and contributed highly constructive suggestions that helped us improve the robustness of our analyses.

While we agree with Reviewer #4 on the importance of novelty and methodological rigor, we respectfully disagree with their specific assessments. We have substantiated our arguments in detailed point-by-point responses below.

Nevertheless, we appreciate Reviewer #4's feedback, which helped us identify areas where our arguments could be made clearer, especially when it comes to the novelty of our work. We have made further improvements accordingly (see response below and the manuscript with tracked changes), building on the substantial efforts already made through several rounds of revisions, both before and after submission, to strengthen the quality of the manuscript.

We believe the revised manuscript is well-suited for *Nature Communications* in terms of importance, novelty, and rigor. We truly and sincerely hope the editor and reviewers will reconsider the potential of this work for publication in *Nature Communications*.

Part II - Point-by-point responses to the comments

Please find a detailed response to all the *reviewer' comments (in Italics)* below, with crucial information highlighted in blue font.

Reviewer #1 (Remarks to the Author):

Comment:

Interesting and relevant study on global hotspots of agricultural conversion, and the environmental impacts of conversion. By identifying these hotspots, focused attention can be placed on conservation measures. The authors did an extremely thorough job of addressing my previous reviewer comments, as well as the comments from two other reviewers.

Response:

We sincerely appreciate your recognition of the extensive efforts we made to revise and improve our manuscript. Your comment that we did “*an extremely thorough job*” in addressing all three reviewers' concerns was especially encouraging. Thank you again for your substantial efforts and valuable comments in both the initial and the second round of reviews.

Reviewer #2 (Remarks to the Author):

Comment:

Thank you for the detailed response to my review. You have answered my questions satisfactorily.

Response:

We sincerely appreciate your acknowledgement of the significance of our study in the initial round of review and the improvements we have made through extensive revisions. Your careful evaluation and constructive suggestions have greatly contributed to strengthening the robustness of our analysis.

Comment:

Two additional comments:

"We would, however, expect the scale of non-forest conversion driven by tree crops to be rather small at the global level, as converting land that naturally carries short vegetation to tree crops or tree plantation is highly input intensive." I would challenge this. For a number of reasons - not least conservation efforts towards forests - grasslands have become less controversial places for land use conversion. Large areas of African savannah are identified for afforestation for instance.

I don't think this makes a material difference to your analysis but perhaps a mention in the text.

Response:

Thank you very much for your thoughtful comments and acknowledging the revised manuscript requires only minor textual adjustments.

We have conducted an additional literature review and agree with your suggestion. We found that, while converting land that naturally carries short vegetation to tree plantations could be input intensive, some non-forest short vegetation is still targeted for tree plantations for various reasons. For example, many non-forest lands in India are viewed as unproductive ‘wastelands’ by the Indian government — a legacy of colonial taxation, and many areas that are natural savannas may be misunderstood as degraded forests ¹. This misclassification makes these non-forest ecosystems prime targets for agriculture, as well as for afforestation and other “restoration” initiatives ^{1,2}. We have clarified and incorporated this issue in the discussion of limitations:

(Page 15 Line 23 - 35) “Our work also highlights the need for consistent definitions, classifications and mapping of natural and cultivated ecosystems. To our best knowledge, no existing global fine-resolution land cover products distinguish cultivated pastures from natural grasslands and tree crops from natural forests simultaneously, partly due to technical challenges. They also have different classification schemes for different natural land covers. This not only complicates quantitative comparisons between different analyses, but also hinders the integration of various land cover and land use data (e.g., merging datasets compiled specifically for pastures with those for all types of land covers). More in-depth assessments and refinements should carefully select the most suitable dataset. For example, for some regions where non-forest ecosystems become prime targets not only for herbaceous cropland but also for tree crops and afforestation (e.g., in India ⁶⁴), datasets that distinguish tree crops (e.g., GLC_FCS30D) would be more suitable for conversion assessments.”

Comment:

"HILDA shows that the conversion from unmanaged grassland/shrubland to pasture/rangeland during 2000-2020 was almost twice the conversion from pasture/rangeland to cropland, indicating that our results might underestimate non-forest conversion driven by total agricultural activities." If this were a robust conclusion it would be very important, focusing attention on (from a conservation perspective) loss of grassland quality as being more significant than conversion. But I note your caution on these figures. You need to decide if the data are strong enough to comment on this.

Response:

Thank you for raising this point. As you have already noted our detailed clarification notes, we understand that you recognize the supplementary analyses based on the HILDA dataset does not affect our analyses. However, we would like to further clarify this for the reference of the editors and other reviewers.

Our analyses are based on the three high-resolution datasets: GlobeLand30, GLCLUC, and CLC_FCS30D, while, as suggested, we also provided an additional comparison with the HILDA dataset in the SI.

On the one hand, we acknowledge that the original statement was inaccurate, but it requires only textual adjustments and does not affect our analyses. It should be revised to state that “not distinguishing cultivated and natural grasslands” may underestimate non-forest conversion, rather than that “our results” might do so. This is because HILDA defines pasture as managed grassland, and GlobeLand30, one of the datasets underpinning our main analyses, does distinguish cultivated/managed and natural grasslands. Moreover, GlobeLand30 uses high-resolution satellite imagery, incorporating crop phenology, field patterns, and expert validation. This comprehensive approach improves the accuracy of its land cover data (see SI Table 1(d)) and, in turn, the robustness of our study. Still, non-forest conversion assessments can further benefit from future improved land cover datasets.

On the other hand, while HILDA similarly shows extensive non-forest conversion, consistent with other three datasets, it may not be robust enough to support more detailed analyses (as explained in the clarification notes for Fig. S6 already), therefore, we only included this specific HILDA-based finding in our previous response letter for your reference and did not use them in our main analyses. These concerns stem from several limitations of HILDA. Most notably, HILDA does not directly use satellite image to identify pasture, instead, it relies on FAO’s Gridded Livestock of the World v3 (2010), which has a much coarser resolution of ~9.28 km (vs. 30 m for the three datasets we used) and covers only a single year. Its overall spatial resolution (1 km) is also coarser than the other three datasets and downscaling the original 9.28 km pasture data to 1 km may introduce additional uncertainty.

Meanwhile, both natural ecosystem conversion (e.g., conversion from natural grassland to cultivated grassland) and quality degradation (e.g., utilization of natural grassland with varying land-use intensities) are critical conservation concerns. This underscores the need for future global land cover products that can capture ecosystem degradation—an area where current datasets fall short due to significant technical challenges. Our study provides a necessary starting point for more comprehensive global analyses on natural non-forest ecosystems.

Reviewer #4 (Remarks to the Author):

Comment:

General Comment.

Before starting my evaluation, I must clarify that this review follows two other evaluation processes initiated by other reviewers, which have improved the quality of the manuscript.

Using secondary information (three general global models of land covers, limits of protected areas, and range-polygons of all the threatened amphibians, mammals, reptiles provided by IUCN), authors of this manuscript sought to 1) globally identify hotspots of conversion from non-forested natural areas to non-natural covers, 2) evaluate this landcover change in protected areas, and 3) estimate the possible effect of this landcover change on the conservation of threatened species.

*In general terms, I found that the methods proposed to develop the research objectives were well executed, which would allow for generating results consistent with used resources. However, my conclusion is that the manuscript should not be published in **Nature** but it can be published in another Springer journal that has lower requirements in relation to quality of the data and novelty. The details of this decision are set out below.*

Response:

First of all, we sincerely appreciate the reviewer's acknowledgement of the improvement we have made through great efforts and extensive revision, as well as the statements that *"the methods proposed to develop the research objectives were well executed, which would allow for generating results consistent with used resources"* and analysis of non-forest conversion to cropland *"would be a major step forward the conservation of these areas on the planet"*. This aligns with the positive comments of all the other three reviewers, who explicitly acknowledged the novelty and robustness of our analyses.

While we agree that papers submitted to *Nature Communications* should be novel and methodologically rigorous, we respectfully disagree with the specific assessments, *as they are either not substantiated by evidence or seemingly based on misunderstandings of our study and the existing literature*. Please see our point-to-point response below.

Nevertheless, we appreciate your valuable feedback, which helped us identify areas where our arguments could be presented more clearly. We have carefully made further revisions according to the detailed comments below. We believe the revised manuscript is a good fit for *Nature Communications*, especially following our careful revisions to enhance clarity based on your feedback.

We have devoted substantial efforts throughout multiple rounds of revision to improve the manuscript. We truly and sincerely hope you could reconsider the possibility of this work for publication in *Nature Communications*.

Comment:

Clearly, conducting a precise and comprehensive analysis of how non-forested natural

areas have been transformed by anthropogenic covers would be a major step forward the conservation of these areas on the planet. However, the main problem that I identified is the low confidence of the landcover maps used for the analysis. It is known that these three landcover global maps (GlobeLand30, Global Land Cover and Land Use Change-GLCLUC, GLC_FCS30D) do not have enough local accuracy in identifying non-tree natural covers, such as savannas, grasslands, and wetlands against non-natural grasslands, such as, non-natural pastures, crops and tree and shrub plantation. Consequently, the results shown in this work have inaccuracies that cannot be controlled by authors.

Response:

We appreciate the reviewer's comment, however, with respect, we note that the reviewer's claim of "*the low confidence of the landcover maps used for the analysis*" is presented without any supporting references. This contrasts with a substantial body of peer-reviewed literature that utilized these land cover products and demonstrated their reliability, including many studies published in leading journals such as *Nature Communications*, *Nature Sustainability* and other Springer journals ³⁻⁶.

In addition, since there are no "perfect" datasets that can support a fully accurate assessment, especially at a global level, to ensure results robustness, we have carefully designed the methods with the "great care taken to choose the land cover datasets and explain their benefits and limitations" (Reviewer #3).

Moreover, by systematically comparing across these datasets, we have also highlighted both their usefulness and limitations, and identified key challenges for such global-scale analyses, offering valuable insights into where improved data are most needed for future studies.

Specific justifications are provided below:

- (1) To eliminate misestimation due to local inaccuracy of a single land cover product, we have compared results across three different high-resolution land cover products. We only highlighted the findings consistently detected by all the three datasets, and clearly and comprehensively discussed the inconsistencies as well as the potential reasons and limitations. Our result shows a high degree of agreement across the three datasets in terms of where conversion hotspots are located geographically, also in contradiction to the reviewers sweeping statement on lack of local accuracy.
- (2) All of the datasets we used adopted different methodologies that greatly improved their accuracy. GLC_FCS30D applied a change detection algorithm ⁷, which is considered one of the best available practices in identifying pixels of change ⁸. GLCLUC applied a four-year epoch to distinguish cropland from natural grassland — assigning the cropland class if cropland was present in any of the four years — which helps reduce misclassification of temporarily fallow land ⁹. It also used

vegetation height for land classification, thereby minimizing misclassification between forest and non-forest natural land cover, as tree height is usually an important standard to define forests¹⁰. GlobeLand30 differentiated cultivated grasslands from natural grasslands based on satellite image, applying a three-step approach that integrates crop and grass phenology (e.g., planting, growth, maturity, and harvest), spatial patterns of cultivated land (e.g., circular or rectangular fields), and expert validation¹¹, which can improve the identification accuracy.

- (3) All three datasets demonstrate high user's and producer's accuracies based on their respective accuracy assessments^{7,9,11}, which we have additionally summarized in the Supplementary Information Table S1 (d). Moreover, as cultivated land/cropland, forestland, and built-up areas are mapped with high precision — reflected in both user's and producer's accuracy — the likelihood of misclassifying these categories as non-forest natural land covers (and vice versa) is likely to be low. Consequently, the accuracy of our conversion assessments is also expected to be high.
- (4) Technical challenges in remote sensing inevitably introduce some uncertainties in the development of land cover datasets, particularly those aiming for global coverage across all land cover types. While region-specific or single land cover-focused datasets may offer higher accuracy for local areas or specific land cover types, their limited scope makes them unsuitable for our globally consistent cross-country and cross-land cover comparisons. Our study represents a necessary starting point for more precise global analyses. Moreover, we identified and highlighted key challenges in assessing land cover conversion at a global scale, and provided valuable insights into where improved data are most needed to better detect non-forest ecosystem conversion.

While we disagree with the reviewer's conclusion, we sincerely appreciate the reviewer raising this point. It has prompted us to further clarify these points:

In the Discussion section, we added the explanation that:

(Page 15 Line 13 - 21) “Technical challenges in remote sensing inevitably introduce some uncertainties in the development of land cover datasets, particularly those aiming for global coverage across all land cover types. While region-specific or single land cover-focused datasets may offer higher accuracy for local areas or specific land cover types, their limited scope makes them unsuitable for our globally consistent cross-country and cross-land cover comparisons. Future studies would benefit from improved land cover products tailored to specific research objectives. Until such products become widely available, incorporating local, context-specific knowledge remains essential for verifying and fully understanding land use dynamics.”

In the Introduction section, we clarified that:

(Page 4 Line 24 - 40) “To minimize potential misestimation from relying on a single data source, the analyses were conducted using three state-of-the-art land cover datasets: GlobeLand30²⁷, Global Land Cover and Land Use Change (GLCLUC)²⁸, GLC_FCS30D²⁹. Each is compiled using different methods to ensure high accuracy, along with different land cover classification schemes (see Supplementary Information (SI)), meaning they can offer complementary information to reveal a more holistic and robust picture of agriculture-driven land conversion. For example, GLC_FCS30D applies a change detection algorithm that enhances accuracy in identifying pixels of change³⁴ and it is the only dataset that includes tree crops – a major driver of agricultural expansion in Southeast Asia and Africa^{35,36} – in its cropland class. Only GlobeLand30 tries to differentiate cultivated pastures from natural grasslands, and it uses a rigorous three-step method to improve accuracy, which is important given the large share of grazing in agricultural land use³⁷. GLCLUC, however, uses 4 years of data for the identification of cropland to minimize misclassifications (e.g., due to fallow). Therefore, the comparison across datasets allows us to account for methodological limitations and misclassifications of the underlying remote sensing products datasets, as well as ensure that the conversion dynamics captured by the analysis are robust.”

Comment:

*At the last part of the introduction, authors suggested that this study can help to generate conservation and planning strategies on non-forest natural vegetation (a point already proposed by other authors using the same used global maps of this work) and clarify that this work did not intend to precisely quantify conversion rates of non-forest natural vegetation. The issue is that we already have other similar analysis (as in this study) that proposed that we are at the point where it is necessary precisely quantify conversion rates of non-forest natural vegetation. The previous show that this work is not a novel analysis at level of **Nature**.*

*Following with the novelty point, authors added in the introduction information on global and regional estimates of conversion, carbon stock and biodiversity in grassland and wetlands, showing that we already have similar global analyses. Also, other authors have already published global analysis on grassland and wetland conversion; therefore, the presented analysis could not be considered as a total novel analysis, which is not negative, but then this would not meet **Nature**'s novelty requirements. The next some other similar global analysis already published:*

- *Annual al 30-m maps of global grassland class and extent (2000–2022) based on spatiotemporal Machine Learning: <https://doi.org/10.1038/s41597-024-04139-6>*
- *Global wetland maps: <https://www2.cifor.org/global-wetlands/>*
- *The Global Lakes and Wetlands Database (GLWD) version 1 and 2 (it includes mapping and analysis between 1990-2020: <https://www.hydrosheds.org/products/glwd>.*
- *Global Wetland Watch: mapping and monitoring changes to wetland ecosystems: <https://unepdhi.org/global-wetland-watch-mapping-and-monitoring-changes-to-wetland-ecosystems/>*

Response:

Thanks for this comment, but we respectfully disagree with the conclusion of “*not a novel analysis at level of Nature*”, as these comments are not substantiated by any or valid evidence and based on misunderstandings of our study and the existing literature. To the best of our knowledge, there are no quantitative assessments of global natural non-forest ecosystem conversion.

Regarding the reviewer’s claim that there are already similar analyses:

- (1) First, no references have been provided for the claim that other authors have already proposed similar points “*using the same used global maps of this work*”.
- (2) Second, all the references provided in the later paragraphs address topics that are **fundamentally different** from the focus of our study. They only show the **distribution** of grasslands and wetlands, **not their conversion** (i.e. no temporal dynamics). As there is no focus on conversion, they also **do not look into how cropland expansion drives changes** — which is the core of our study.

Additionally, most of these references are just online land cover datasets, not peer-reviewed analytical studies published in academic journals.

- (3) Third, the reviewer also misunderstood the references we cited in the Introduction. The literature on non-forest ecosystem conversion is **local-scale** and **mostly qualitative** studies, **limited to a very few and small regions** (e.g., the state of Mato Grosso in Brazil and the Cerrado), which further underscores the necessity and novelty of **our first global-scale quantitative** analysis. The figures we cited regarding carbon and biodiversity in grasslands and wetlands are based on their total **distribution, not conversion**.

The comments made us realize that the purpose of references in the intro was not stated as clearly as it could be, we have therefore further clarified the different scopes and focuses of our study and existing literature:

(Page 3 Line 26 - 30) “While there are emerging efforts to map the distribution of non-forest ecosystems^{18,19}, quantitative analyses of their conversion remain limited to only a few regions (e.g., the Cerrado¹³), and no study to date has specifically examined their conversion dynamics at a global scale.”

- (4) Fourth, our analysis not only studied non-forest ecosystem conversion, but also examined conversion within **protected areas** and assessing implications for **biodiversity**, which have important academic and policy implications and further strengthened the novelty of our study.

Given these, we strongly agree with Reviewer #1-3 that this study meets the novelty requirements of *Nature Communications*.

Regarding the reviewer's claim that it is necessary to precisely quantify conversion rates of non-forest natural vegetation:

While our main focus is identifying conversion hotspots, our study is fundamentally based on quantitative assessments of conversion rates, as demonstrated by the statistical analyses and figures in the Results section.

The comments helped us recognize that our point was not stated as clearly as intended. We have accordingly revised the text to improve clarity:

(Page 5 Line 1 - 8) "By identifying global conversion hotspots, this study can assist companies and public policymakers in the design and prioritization of sustainable land-use policies, such as future conservation planning. Moreover, our first quantitative global-scale assessment of conversion rates as well as the cross-dataset comparisons not only provide a starting point for future improved analyses of land use transitions and associated carbon and biodiversity impacts, but also advances our understanding of the limitations of currently available datasets and challenges for land conversion assessments, offering valuable insights into where improved data are most needed to support more robust analyses."

Comment:

*Based on the criteria established by **Nature** for its publications, my conclusion is that this work does not meet the standards of precision and novelty that a publication in Nature must have to. I realize we are at the point where we need the results of precise analyses focused on non-forested natural areas and not what this work does; this work reanalyzed general global maps where the classification error of non-forested areas is too high locally and/or cannot be controlled by the authors. Authors' conclusion is obvious: non-forested natural areas have decreased over time, this work would be enough 10 years ago, but we need more now. We now need more than obvious general trends. We need spatial details about how these trends change within country regions, which are the analyses that this paper doesn't show. My recommendation is that the author look for a lower impact factor journal to submit.*

Response:

Thanks for the comment. The general concerns about analysis robustness and novelty have been explained in the responses above. We believe that this manuscript meets the standards of ***Nature Communications*** in terms of importance, novelty, and rigor.

Regarding the specific comment on spatial details:

We agree with the reviewer that detailed statistics are necessary for targeted policymaking, but the comment of lack of spatial details appears to overlook the fact that we provide not only general trends of global natural non-forest conversion, but also conversion at the pixel level (see Fig. 1a and Fig. S1) and at the country level (Fig. 1c). For your convenience, Fig. 1a and Fig. 1c are included below.

Editorial Note: Administrative boundaries were modified from GADM version 4.1 in Fig. 1a.

Fig. 1a Global distribution of non-forest conversion hotspots in 2000-2020. Values (mean from the 3 datasets) are presented as the percentage of 30 m by 30 m pixels in a 10 km by 10 km grid cell.

Fig. 1c Conversion of grassland & shrubland (marked in light grey) and wetland (marked in dark purple) for the 5 hotspot countries/regions appeared in all 3 datasets and the 7 hotspot countries/regions appeared in any 2 of the datasets.

Moreover, we have produced more detailed data, but considering that this study is the first attempt to quantify global-scale non-forest ecosystem conversion and that we also aim to reveal conversion within protected areas and the impacts on biodiversity, we choose to retain only the most essential information in the main text due to space constraints. However, as suggested, we have now additionally provided time series analyses at finer scales in the Supplementary Information Fig. S2. The figure is also pasted below for your reference.

Editorial Note: Administrative boundaries were modified from GADM version 4.1 in Fig. S2.

Fig. S2 Decrease (–) and increase (+) in conversion during 2010–2020 compared to 2000–2010. Only areas where all three datasets show conversion and their average falls within the top 90% of global total conversion are colored. Values are represented as the average change estimated across the three datasets.

Comment:

Additional specific comments

Pag6. Why did authors provide a range for forest cover and non-forest natural land covers? (At a global scale, there was extensive and increasing non-forest natural land covers spanned 3.0 – 5.5 Gha in 2000 while forest areas covered 4.1 – 5.1 Gha). It means that the estimates of the three used maps are highly contrasting and would be very inaccurate. So again, we're at a point where the analysis presented is correct, but we need to use more accurate maps. Thus, we need these types of maps first.

Pag7. This result is not expected (The variations across the three datasets for non-forest conversion is much smaller than that for forest conversion). In terms of remote sensing, the classification of forests is easier than other vegetation (grassland, wetlands crops, etc.) because covering, photosynthetic activity, and structure are more stable in forest than crops and non-forest natural land covers.

Response:

Thanks for raising this point, but the comment overlooked that the differing classification schemes used across the three datasets (including definitions of forests), can contribute to the observed statistical discrepancies. We have made this point clearer in the revised version.

As suggested by Reviewer #2 and #3, we have made great efforts to clarify their differing classification schemes (outlined in Table 1 and Tables S1–S3) and comprehensively discussed the potential implications of the statistical discrepancies (the 4th and 5th paragraphs of Results section). For instance, regarding the reviewer’s concern on forests, GLC_FCS30D sets lower tree canopy threshold for forests than GlobeLand30, and neither GlobeLand30 nor GLC_FCS30D specify whether tree height was a criterion for forests, while GLCLUC does not consider tree canopy information, both key criteria for FAO’s definition of forests. Moreover, GlobeLand30 and GLC_FCS30D classify Tundra as a separate category but use different classification standards, whereas GLCLUC includes it within tree cover and short vegetation. **Despite these discrepancies, all the datasets consistently show common conversion hotspots at both the pixel and country levels (see Fig. 1).**

Moreover, in addition to presenting the first global quantitative analysis of non-forest ecosystem conversion, another contribution of this study is to reveal limitations of existing global land cover products through such comparison and identifying priority areas for future improvement, as clarified in the revised *Introduction* and *Discussion* section. For instance, our findings underscore the urgent need for improved identification as well as consistent land cover definitions and classifications in remote sensing — particularly the clear delineation between forest and non-forest, and between natural and cultivated ecosystems. Currently, the differing classification schemes adopted by existing products hampers cross-dataset comparisons and the integration of land cover and land use information (e.g., merging pasture-specific datasets with broader land cover maps). Until higher-accuracy datasets with more detailed and consistent classifications become available, future assessments must carefully select the most appropriate dataset. For example, in regions where tree plantations are the main driver, datasets that distinguish tree crops (e.g., GLC_FCS30D) are more appropriate, while in regions where pasture expansion is the main driver, datasets that differentiate pastures (e.g., GlobeLand30) are more suitable.

Reference:

1. Lahiri, S. & Reddy, S. India’s grasslands are not “wastelands”. *Science* 387, 726–727 (2025).
2. Stevens, N., Bond, W., Feurdean, A. & Lehmann, C. E. R. Grassy Ecosystems in the Anthropocene. *Annual Review of Environment and Resources* 47, 261–289 (2022).
3. Winkler, K., Fuchs, R., Rounsevell, M. & Herold, M. Global land use changes are four times greater than previously estimated. *Nat. Commun.* 12, 2501 (2021).
4. Han, J. et al. Threat of low-frequency high-intensity floods to global cropland and crop yields. *Nat. Sustain.* 7, 994–1006 (2024).
5. Parente, L. et al. Annual 30-m maps of global grassland class and extent (2000–2022) based on spatiotemporal Machine Learning. *Sci. Data* 11, 1303 (2024).
6. Chen, B. et al. Wildfire risk for global wildland–urban interface areas. *Nat. Sustain.* 7, 474–484 (2024).
7. Zhang, X. et al. GLC_FCS30D: the first global 30m land-cover dynamics monitoring product with a fine classification system for the period from 1985 to 2022 generated using dense-time-series Landsat imagery and the continuous change-detection method. *Earth System Science Data* 16, 1353–1381 (2024).

8. Olofsson, P. et al. Good practices for estimating area and assessing accuracy of land change. *Remote Sensing of Environment* 148, 42–57 (2014).
9. Potapov, P. et al. The Global 2000-2020 Land Cover and Land Use Change Dataset Derived From the Landsat Archive: First Results. *Front. Remote Sens.* 3, (2022).
10. FAO. *Global Forest Resources Assessment 2010 - Terms and Definition.* (2010).
11. Chen, J. et al. Global land cover mapping at 30 m resolution: A POK-based operational approach. *ISPRS Journal of Photogrammetry and Remote Sensing* 103, 7–27 (2015).

Response to the comments

Reviewer #4 (Remarks to the Author):

Comment:

As I concluded previously, all methods proposed to develop the research objectives were well executed, which allowed for generating results consistent with used resources. As well, it is valuable that authors have worked and improved all the minor comments that I suggested in my first revision. Also, I have evaluated carefully the responses of authors to my two main criticisms to the manuscript: 1) the three landcover maps used by authors have inaccuracies that authors cannot control and 2) manuscript does not meet the standards novelty that a publication in Nature must have to.

Regarding the first criticism, authors have provided all the evidence to show that these global land cover maps are the best models we have for assessing the loss of non-forested natural areas, and they emphasized that any remote sensing analysis (such as these three novel maps and their respective publications) is inherently flawed. The authors explain that they linked the three maps to increase the confidence of their analysis, and I agree with this. My conclusion regarding this first criticism is that, by combining the information from the three maps, the authors have some control over, improving the confidence of the maps, thus providing certainty to the analysis in this manuscript. Therefore, I concluded this point is resolved

Regarding the second criticism, I emphasize that this manuscript is a reanalysis of three published models of land covers (where each one was novel analysis by the time its publication) and other published species distribution models. Therefore, I believe this manuscript does not constitute a novel analysis. However, I reviewed Nature's publication standards, and it is not clear whether a reanalysis of previously published models constitutes a novel analysis. I personally conclude that the analysis is not novel, consequently I suggest the article should be published in a lower-impact journal. However, I prefer that the decision be made by the editor in chief.

Response:

First of all, we sincerely appreciate the reviewer's careful re-evaluation of our revised manuscript and the detailed point-by-point responses to their previous concerns. We are also pleased to note the reviewer is satisfied that we "have worked and improved all the minor comments" and the issues regarding methodological robustness have been resolved.

With respect to the concern on novelty, we continue to maintain that our study meets the novelty requirements, which is consistent with the views of other reviewers, such as "Interesting study on a topic that does not receive much attention" (Reviewer 1) and "Thank you for an interesting publication. There is definitely a need to focus on non-forest ecosystems" (Reviewer 2).

The reviewer's concern regarding novelty is that our analysis relies on existing datasets. However, we believe that in-depth investigation of an important scientific question is equally valuable as the development of new datasets, which on their own do not provide analytical insights. Moreover, our study addresses a fundamentally different question from these datasets: they only focus on the distribution of different land covers, but do not address key research questions, such as the dynamics and patterns of non-forest ecosystem conversion (beyond static distribution), the role of agricultural activities in driving these changes, or how such evidence can inform companies and policymakers in designing sustainable land-use policies and integrated policy frameworks that avoid trade-offs and advance broad sustainability goals.

We would also like to reiterate that, our analysis not only studied non-forest ecosystem conversion, but also (1) assessed conversion within protected areas and associated biodiversity impacts, which have important implications for ecosystem conservation strategies, and (2) comprehensively discussed the usefulness and limitations of existing datasets and key challenges for such global-scale analyses, which offer valuable insights into where improved data are most needed for future studies. These contributions further strengthen the novelty of our study.

Besides, we have included a more explicit statement about where this manuscript sits within the existing work (Page 2 Line 18-24):

“Although a few local-scale studies have shown that non-forest ecosystems are critically imperiled, most existing literature have focused either on mapping distribution of non-forest lands ^{12,13} or on analyzing forest conversion ³⁻⁵. To date, however, no study has comprehensively assessed the global dynamics and impacts of non-forest ecosystem conversion (beyond static distribution), and particularly, the role of agricultural expansion in driving these changes, which represents an urgent gap that must be addressed to inform sustainable land use policies.”

We hope that these clarifications adequately address the reviewer's concerns and further demonstrate the novelty and significance of our work.